# Rarγ-Foxa1 signaling promotes luminal identity in prostate progenitors and is disrupted in prostate cancer

Dario De Felice [1,7], Alessandro Alaimo [1,7], Davide Bressan [1,7], Sacha Genovesi [1], Elisa Marmocchi[1], Nicole Annesi[1], Giulia Beccaceci [1], Davide Dalfovo [1], Federico Cutrupi[1], Stefano Medaglia[1], Veronica Foletto [1], Marco Lorenzoni [1], Francesco Gandolfi[1], Srinivasaraghavan Kannan[2], Chandra S Verma[2,3,4], Alessandro Vasciaveo[5], Michael M Shen [5], Alessandro Romanel [1], Fulvio Chiacchiera[1], Francesco Cambuli [1,6,8✉] & Andrea Lunardi [1,8✉]

## Abstract

Retinoic acid (RA) signaling is a master regulator of vertebrate development with crucial roles in body axis orientation and tissue differentiation, including in the reproductive system. However, a mechanistic understanding of how RA signaling governs cell lineage identity is often missing. Here, leveraging prostate organoid technology, we show that RA signaling orchestrates the commitment of adult mouse prostate progenitors to glandular identity, epithelial barrier integrity, and specification of prostatic lumen. RA-dependent RARγ activation promotes the expression of Foxa1, which synergizes with the androgen pathway for luminal expansion, cytoarchitecture and function. *FOXA1* mutations are common in prostate and breast cancers, though their pathogenic mechanism is incompletely understood. Combining functional genetics with structural modeling of FOXA1 folding and chromatin binding analyses, we discover that FOXA1$^{F254E255}$ is a loss-of-function mutation compromising its transcriptional function and luminal fate commitment of prostate progenitors. Overall, we define RA as an instructive signal for glandular identity in adult prostate progenitors. Importantly, we identify cancer-associated FOXA1 indels affecting residue F254 as loss-of-function mutations promoting dedifferentiation of adult prostate progenitors.

**Keywords** Prostate; Organoids; FOXA1; Retinoic Acid
**Subject Categories** Cancer; Development; Stem Cells & Regenerative Medicine

## Introduction

The extensive self-renewal potential of the prostate tissue was demonstrated more than 40 years ago, through elegant androgen cycling experiments performed in rodents, showing that the adult prostate epithelium was capable of surviving castrate levels of testosterone (T) and retaining ample regenerative potential after androgens replenishment (Evans and Chandler, 1987; Evans, 1988). However, the widespread cellular quiescence typical of the adult prostate epithelium, coupled with rapid adaptation to in vivo and ex vivo perturbations, has complicated the identification and characterization of stem/progenitor cell populations and their niches. Early studies observed "*intermediate*" progenitor cells expressing both basal and luminal markers (Wang et al, 2001). Afterward, cell transplantation experiments and lineage tracing studies in animal models showed that both basal and luminal compartments contain multipotent progenitors characterized by remarkable plasticity and capable of differentiating towards both luminal and basal cell lineages (Lawson et al, 2007; Wang et al, 2009; Chua et al, 2014; Wuidart et al, 2016). Lately, single-cell transcriptomic studies have provided a more granular view of cell identities in the adult prostatic epithelium, revealing the proximal anatomical district as the preferred niche for a variety of epithelial progenitor cells (basal, luminal proximal—LumP and periurethral—PrU), although they are also unfrequently found in the distal compartment (Crowley et al, 2020; Guo et al, 2020; Henry et al, 2018; Joseph et al, 2020; Karthaus et al, 2020; Mevel et al, 2020). These progenitor cells, which are largely dormant in adult tissue (Kwon et al, 2016; Pignon et al, 2013), display great regenerative potential in ex-vivo assays (Crowley et al, 2020; Kwon et al, 2016).

At the molecular level, a subset of key transcription factors, including *NKX3.1*, *FOXA1*, *HOXB13*, *SOX9*, *AR*, and *TRP63* genes has been identified as major determinants of prostate epithelium

[1]Department of Cellular, Computational and Integrative Biology (CIBIO), University of Trento, 38123 Trento, TN, Italy. [2]Bioinformatics Institute (Agency for Science, Technology and Research, A*STAR), 30 Biopolis Street, 07-01 Matrix, Singapore 138671, Singapore. [3]Department of Biological Sciences, National University of Singapore, 14 Science Drive, Singapore 117543, Singapore. [4]School of Biological Sciences, Nanyang Technological University, 60 Nanyang Drive, Singapore 637551, Singapore. [5]Departments of Medicine, Genetics & Development, Urology and Systems Biology, Herbert Irving Comprehensive Cancer Center, Columbia University Irving Medical Center, New York, NY 10032, USA. [6]Present address: Human Technopole, via Rita Levi Montalcini 1, Milan, Italy. [7]These authors contributed equally as first authors: Dario De Felice, Alessandro Alaimo, Davide Bressan. [8]These authors jointly supervised this work: Francesco Cambuli, Andrea Lunardi. ✉E-mail: francesco.cambuli@fht.org; andrea.lunardi@unitn.it

differentiation and morphogenesis (Toivanen and Shen, 2017). Among them, the androgen receptor (AR) is the most studied and best characterized signaling transducer and transcription factor acting in the prostate tissue, due to its relevance for reproductive biology and prostate cancer. Beyond AR, limited evidence connects specific signaling pathways with transcriptional regulators and the control of prostate epithelium differentiation and homeostasis (Kruithof-de Julio et al, 2013; Cambuli et al, 2022; Lorenzoni et al, 2022). While the dissection of signaling network in vivo is complicated by the multiplicity and multidirectional nature of ligand-receptor interactions, there are cases where classical genetic studies in animal models have provided undeniable evidence for their relevance. In 1993, Pierre Chambon described *Rary*-null males as sterile due to squamous, instead of secretory, differentiation of the prostate epithelium and seminal vesicles (Lohnes et al, 1993), thus unveiling a critical function of retinoic signaling in the establishment of the luminal compartment of mouse prostate. Retinoids are vitamers of Vitamin A involved in many biochemical processes, including cell differentiation and embryonic development (Petkovich and Chambon, 2022). The pleiotropic actions of retinoids are mediated by two families of nuclear receptors: retinoic acid receptors (RARs) α, β, and γ and retinoid X receptors (RXRs) (Heyman et al, 1992; Levin et al, 1992; Allenby et al, 1993). Upon RA binding, these receptors recognize complementary DNA elements in the regulatory regions of selected genes and modulate their expression (Evans, 1988; Beato, 1989).

Specifically, RARγ has been shown to be deregulated in prostate cancer and to influence the androgen receptor cistrome in malignant cell lines (Long et al, 2019; Petrie et al, 2020; Bhowmick and Bhowmick, 2022; Yu et al, 2022; Wani et al, 2023). Yet, understanding the impact of specific signaling events on prostate self-renewal and differentiation requires control over microenvironmental conditions and genetic perturbations, which is more difficult to achieve in conventional cancer cell lines adapted to grow in serum-rich media.

Here, leveraging organoid technology for the study of the prostate, we identify a new molecular link between retinoic acid (RA), its transcriptional mediator RARγ, and the pioneer transcription factor Foxa1, acting in prostate progenitors to enforce glandular identity in cooperation with androgen signaling and its nuclear receptor Ar. Relying on our improved prostate organoid model with enhanced luminal identity, we demonstrated its utility in a preclinical setting, by characterizing the function of recurrent FOXA1 oncogenic mutant isoforms and demonstrating that the most frequent coding alteration, FOXA1$^{F254E255}$, is a hypomorphic variant with reduced chromatin binding, associated with progenitor dedifferentiation and loss of glandular identity. All-trans RA (ATRA) and RARγ agonists can boost FOXA1 expression and mitigate progenitor dedifferentiation. Our study paves the way for pharmacological strategies aimed to restore near-physiological FOXA1 activity in cancer cells.

# Results

## Retinoic acid promotes prostate-like cytoarchitecture and lumen formation in prostate organoids cooperating with androgen signaling

Organoid cultures were established from the prostate of inbred C57BL/6J and outbred CD1 mice based on the protocol originally described by Drost and colleagues (Drost et al, 2016). Briefly, small tissue fragments were enzymatically and mechanically digested, embedded into hydrogel droplets, and cultured in a defined medium including EGF, Noggin, R-spondin 1, the TGF-β inhibitor A83-01, and dihydrotestosterone (DHT) (hereafter ENRAD media, Fig. 1A). Under such growth conditions, primary adult prostate cells proliferate rapidly forming 3D structures primarily made up of a compact and disorganized mass of cytokeratin 5 (Krt5)-positive basal cells intermixed with a small number of ectopically located cytokeratin 8 (Krt8)-positive cells (Fig. 1B,C). Only a small fraction of the 3D structures showed a prostate-like cytoarchitecture displaying a double layer of basal and luminal cells correctly organized to form a lumen (hereafter mouse prostate organoids, mPrOs) (Cambuli et al, 2022; Lorenzoni et al, 2022) (Fig. 1B,C).

Considering the low efficiency in generating prostate-like organoids, we reviewed the literature and identified retinoic acid as a putative differentiation signal in the prostate, based on early work by Pierre Chambon and colleagues on mouse genetic mutants (Lohnes et al, 1993). Nanomolar concentrations of retinol, a Vitamin A derivative that can be metabolized to retinoic acid (RA) (Fig. EV1A), are present in the B-27 supplement (Hore et al, 2016) commonly employed for the growth of primary cells and organoids in vitro (Karthaus et al, 2014; Gao et al, 2014). The replacement of the standard B-27 supplement (hereafter B-27) with the vitamin A-free formulation (hereafter B-27 *zero*) resulted in the loss of the small percentage of large hollow mPrOs, leaving only compact spheroids in culture (Fig. 1D). Conversely, administration of B-27 *zero* complemented with concentrations between 6 and 16 nM of all-trans retinoic acid (ATRA) (hereafter B-27 *plus* or ENRAD<u>A</u>) significantly boosted RA signaling (as demonstrated by activation of well-characterized target genes as *Aldh1a1* and *Rarb*) and enhanced the formation of mPrOs (Fig. 1D,E,G,H and EV1B–D) characterized by juxtaposed basal (Krt5) and luminal (Krt8) cells surrounding a well-defined expanded lumen (Cambuli et al, 2022; Lorenzoni et al, 2022) (Fig. 1F). We observed proper polarization of luminal cells and epithelial barrier integrity, as visualized by immunostaining for apical (e.g., Zo1 and 3) and junctional markers (e.g., Cldn4 and Cldn7) (Garcia et al, 2018) (Fig. 1F). Notably, RA signaling was necessary but not sufficient to promote prostate-like cytoarchitecture and lumen formation in prostate organoids and acted in concert with the androgen pathway (Figs. 1F–H and EV1D). Similar to androgens (Cambuli et al, 2022), retinoic acid signaling displayed a fully reversible phenotypic switch in culture, as shown by cycling experiments (Fig. 1G,H).

## A Rary-Foxa1 transcriptional cascade is essential for the retinoic acid control of luminal identity in adult prostate progenitors

To understand the impact of retinoic acid signaling on prostate progenitors at the molecular level, we performed bulk RNA-seq on mPrOs grown in the absence or presence of DHT and ATRA for 6 days (ENRA-- (neither DHT nor ATRA) *vs*. ENRAD- (DHT only) *vs*. ENRA-A (ATRA only) *vs*. ENRADA (both ATRA and DHT), *n* = 3 *per* condition).

Focusing on genes displaying at least 2-fold variance in the mean expression (adj. *p*-value < 0.05) across pairwise comparisons, we observed that DHT supplementation led to 459 DEGs (2.6% out of 17750 detectable coding genes, 123 UP/336 DOWN), which

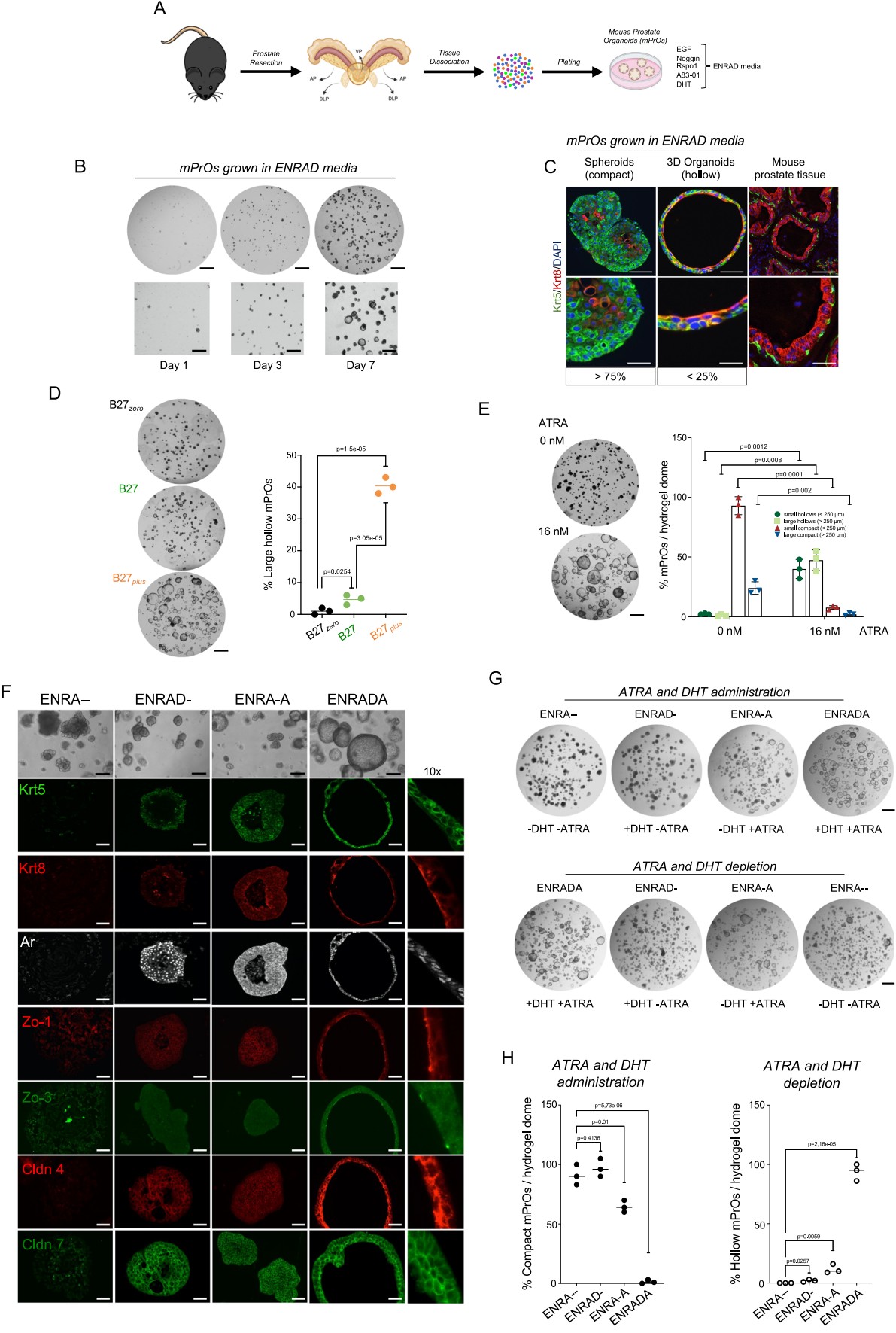

**Figure 1.  Retinoic acid promotes prostate-like cytoarchitecture and lumen formation in prostate organoids cooperating with androgen signaling.**

(**A**) Schematic overview of the procedure for establishing mPrOs (adapted from Karthaus et al, 2014). (**B**) Representative stereoscopic images of the mixed organoid population at different days of culture, scale bar: 1 mm. Magnifications (2x) are shown in the lower panels. Scale bars, 500 μm. $N > 3$ independent biological replicates. (**C**) Immunofluorescence staining of basal (Krt5) and luminal (Krt8) cytokeratins in mPrOs and mouse prostate tissue. Cell nuclei are stained with DAPI. Scale bars, mPrOs 50 μm; prostate 500 μm. Magnification of the selected area are shown. Scale bars, mPrOs 20 μm, prostate 200 μm. $N > 3$ independent biological replicates. (**D**) Representative stereoscopic images (left) and quantification (right) of mPrOs cultured in medium conditioned with different concentrations of ATRA (B27$_{zero}$ = 0 nM, B27 = 6 nM, B27$_{plus}$ = 16 nM; a minimum of 100 organoids/dome x 3 domes were counted for each condition; large mPrOs, diameter >250 μm). Scale bar, 1 mm. Data are presented as mean value ± s.d. of $n = 3$ independent biological replicates. Unpaired t-test, $p < 0.05$ was considered statistically significant. (**E**) Representative stereoscopic images (left) and quantitative phenotypic comparison (right) of mPrOs cultured with or without ATRA. Scale bar, 1 mm. Data are presented as mean value ± s.d. of $n = 3$ independent biological replicates. Paired t-test, $p < 0.05$ was considered statistically significant. (**F**) Representative stereoscopic images and immunofluorescence analysis of Krt5, Krt8, Ar, Zo-1 (Tjp1), Zo-3 (Tjp3), Cldn4, and Cldn7 expression and localization in mPrOs cultured with or without DHT, ATRA or both. Scale bars, 100 μm; $n = 2$ independent biological replicates. Magnification (10x) of immunostaining of mPrOs cultured in presence of DHT and ATRA (ENRADA medium) are shown to pointing out protein localization. (**G, H**) Representative stereoscopic images (**G**) and quantitative analysis (**H**) of mPrOs morphology upon administration, or withdrawal, of DHT and ATRA. Scale bars, 1 mm. Data are presented as mean value ± s.d. of $n = 3$ independent biological replicates. Unpaired t-test, $p < 0.05$ was considered statistically significant. Source data are available online for this figure.

nearly tripled in the presence of ATRA up to 1307 DEGs (7.4%; 706 UP/601 DOWN). Retinoic acid signaling caused a more extensive perturbation of the mPrOs transcriptome, resulting in ~3000 DEGs (~17%; ~1700 UP/~1300 DOWN), and a relatively lower variance associated with the co-presence of testosterone in the medium (Fig. EV2A). Gene ontology analysis of the DEG subsets revealed that the GO term '*Epithelial Cell Differentiation*' was the most significantly enriched upon ATRA supplementation in culture (Appendix Fig. S1 and Dataset EV1).

Shifting our analysis at the individual gene level to prioritize mechanisms potentially responsible for the retinoic acid-driven transcriptional program, we found that *Foxa1*, a well-known pioneer transcriptional factor associated with prostate luminal identity, was listed among the top upregulated DEGs across all pairwise comparisons including ATRA supplementation, regardless of DHT presence (Figs. 2A–E and EV2B,C). In conjunction with *Foxa1* upregulation, we observed increased expression of the *Androgen Receptor* (*Ar*) and, conversely, downregulation of *Trp63*, which represent key transcription factors for the luminal and basal cell lineages, respectively (Figs. 2D and EV2D). Expanding our analysis to additional markers of basal and luminal cells, we found that ATRA prominently enhanced luminal markers, and especially luminal progenitor markers (e.g., *Krt4, Krt7, Clu, Wfdc2,* and *Ppp1r1b*) (Cambuli et al, 2022; Crowley et al, 2020; Karthaus et al, 2020), with the highest levels observed upon supplementation of both DHT and ATRA (ENRADA) (Fig. EV2D,E). Genes encoding for tight junctions' proteins were also induced by retinoic acid (e.g., *Tjp1* and *Tjp3*, *Ocln*, *Cldn4*, and *Cldn7*) (Fig. EV2E). Conversely, the absence of ATRA (ENRA-- and ENRAD- in B27 *zero*) favored phenotypic and molecular features typical of stratified squamous epithelia, including the expression of late cornified envelope family genes (e.g., *Lce1e*, *Lc1f*, *Lc3d*, *Lcd3e*, and *Lcd3f*) and the formation of spheroids almost entirely made up of basal cells enclosing anucleated cornified cells (Fig. EV2F,G).

Retinoic acid supplementation markedly increased Foxa1 protein expression, which was independent from androgen signaling and Ar transcriptional activity (as demonstrated by enzalutamide treatment) (Figs. 2B–D and EV3A,B). To gain mechanistic insights into the control of prostate organoid cytoarchitecture and Foxa1 expression by retinoic acid signaling, we targeted retinoic acid receptors (Rar*s*) with isoform-specific or a pan-Rar inhibitors. We found that Rarγ inhibition (Rarγ-i) significantly reduced lumen formation as well as Foxa1 mRNA and protein levels in prostate organoids (Fig. 3A–C). Leveraging publicly available datasets (Data ref: Crowley et al, 2020),

we mapped *RARγ* expression in the normal mouse and human prostate epithelium at the single-cell level. We found that *RARγ* is predominantly expressed by progenitor cells in vivo, being highly transcribed by nearly 20% of basal and 30% of luminal proximal progenitors in human, and by 20% of basal, 50% of LumP and 40% of PrU in the mouse (Figs. 3D and EV3C; Appendix Fig. S2). Crucially, Foxa1 is essential for mediating the control of retinoic acid on glandular identity in prostate progenitors. Foxa1 knock-down abolished the ability of RA to generate prostate organoids that have a luminal cavity (Fig. 3E,F). Conversely, constitutive expression of a transgene encoding for Foxa1 leads to luminal priming even in the absence of ATRA and DHT (ENRA-- conditions) and to the formation of large hollow mPrOs with a well-structured prostate-like luminal compartment if androgens are added (ENRAD-) to retinoic acid-depleted media (Figs. 3G–J and EV3D,E).

## Foxa1 occupies enhancers and promoters of key luminal progenitor genes and reshapes genome-wide androgen receptor binding

To shed light on how Foxa1 and Ar transcription factors coordinately promote a luminal progenitor gene expression program and a glandular phenotype in prostate progenitors, we combined our RNA-seq analysis of mouse prostate organoids (mPrOs) grown with or without ATRA and DHT with publicly available Foxa1 and Ar ChIP-seq datasets from mPrOs constitutively expressing a Foxa1 transgene or a control empty vector (EV) (Adams et al, 2019) (Fig. 4A).

In standard organoid culture conditions, RA signaling is limited (retinol <10 nM) and Foxa1 levels are low. Transgenic expression of a lentiviral vector encoding for *Foxa1* mimics ATRA treatment leading to roughly a four-fold increase in Foxa1 expression in comparison to the empty vector control (Adams et al, 2019) (Fig. 2G,H). Reanalysis of Foxa1 ChIP-seq in mPrOs$^{Foxa1}$ *vs.* mPrOs$^{EV}$ led to the detection of more than 7000 Foxa1-bound (F1) distal regulatory elements (F1-DE) and almost 2000 promoters (F1-PE), consistent with the known pioneer ability of this transcription factor (Zaret et al, 2008) (Fig. 4B–F and Dataset EV2). The intersection of the ChIP-seq datasets for Foxa1 with the list of genes upregulated in expression in organoids treated with ATRA (ENRADA) compared to regular medium (ENRAD-) (Fig. EV4A and Dataset EV3) highlighted key luminal and intermediate prostate progenitor markers (e.g., *Krt7, Ppp1r1b, Plaur*) (Cambuli et al, 2022; Crowley et al, 2020; Karthaus et al, 2020),

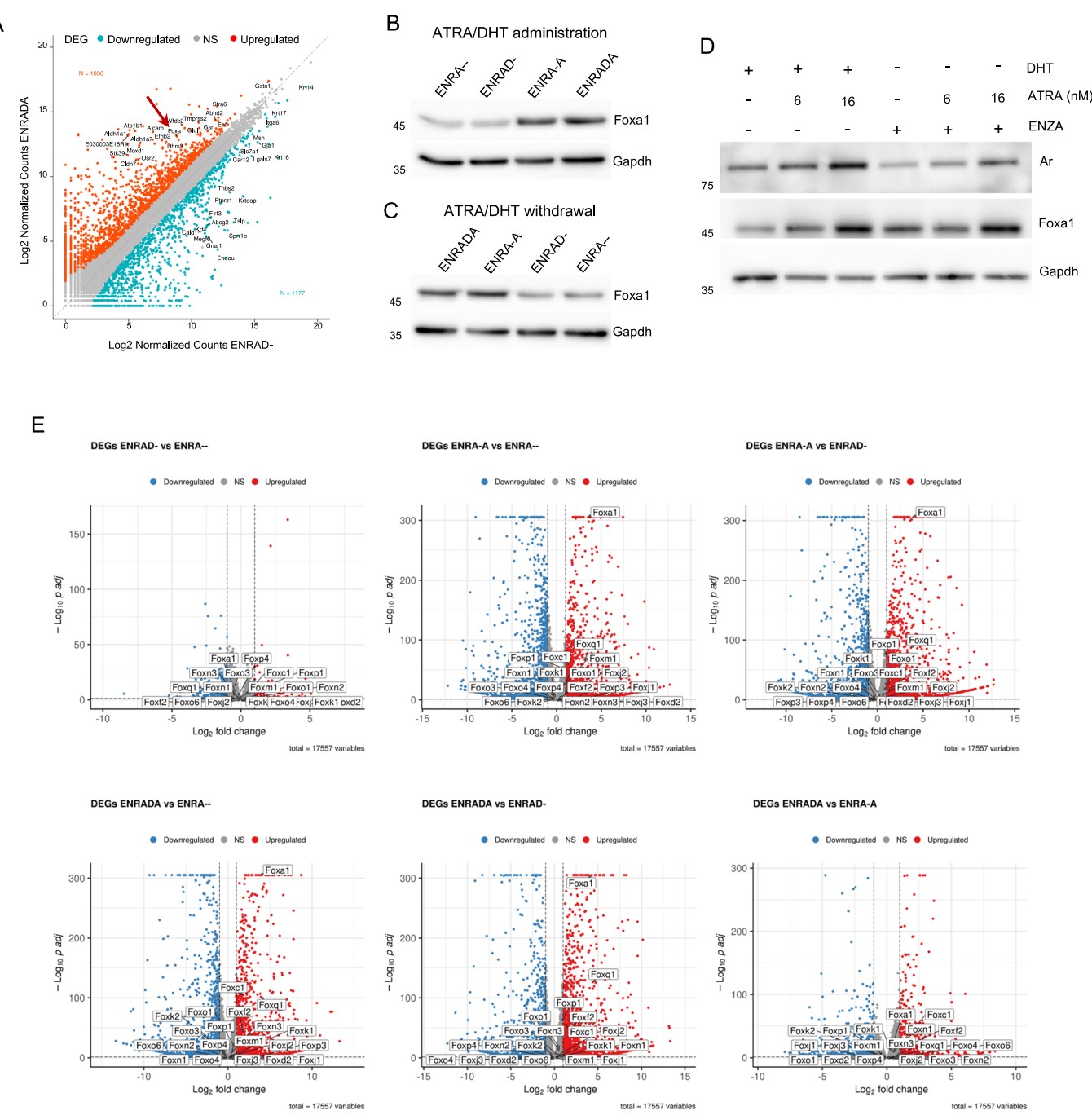

**Figure 2. Retinoic acid signaling promotes Foxa1 expression in adult prostate progenitors.**

(A) Scatter plot representing the changes in gene expressions in mPrOs grown for 6 days in ENRAD or ENRADA conditions. The number of significant up (log2FC > 1, orange) and down (log2FC < −1, light blue) regulated genes is indicated as N in the figure. Significance is assigned if the gene has an adj. *p*-value lower than 0.05 (Wald test followed by the Benjamini–Hochberg multiple test correction, default in DESeq2). Red arrow indicates Foxa1. (B, C) Representative Western blot analysis of Foxa1 expression in mPrOs upon administration for 6 days (B) and successive withdrawal for 6 days (C) of ATRA and DHT individually or in combination. Gapdh is used as loading control. N = 3 independent biological replicates. (D) Representative Western blot analysis of Ar and Foxa1 expression in mPrOs grown for 6 days with or without DHT (10 nM), ATRA (at different concentrations), and Enzalutamide (ENZA, 10 μM). Gapdh is used as loading control. N = 3 independent biological replicates. (E) Volcano plots representing the changes in gene expressions in mPrOs grown for 6 days in indicated media. Members of the Fox family of transcription factors are indicated among the significant up (orange) and down (light blue) regulated genes. Significance is assigned if the gene has an adj. *p*-value lower than 0.05 (Wald test followed by the Benjamini–Hochberg multiple test correction, default in DESeq2). Source data are available online for this figure.

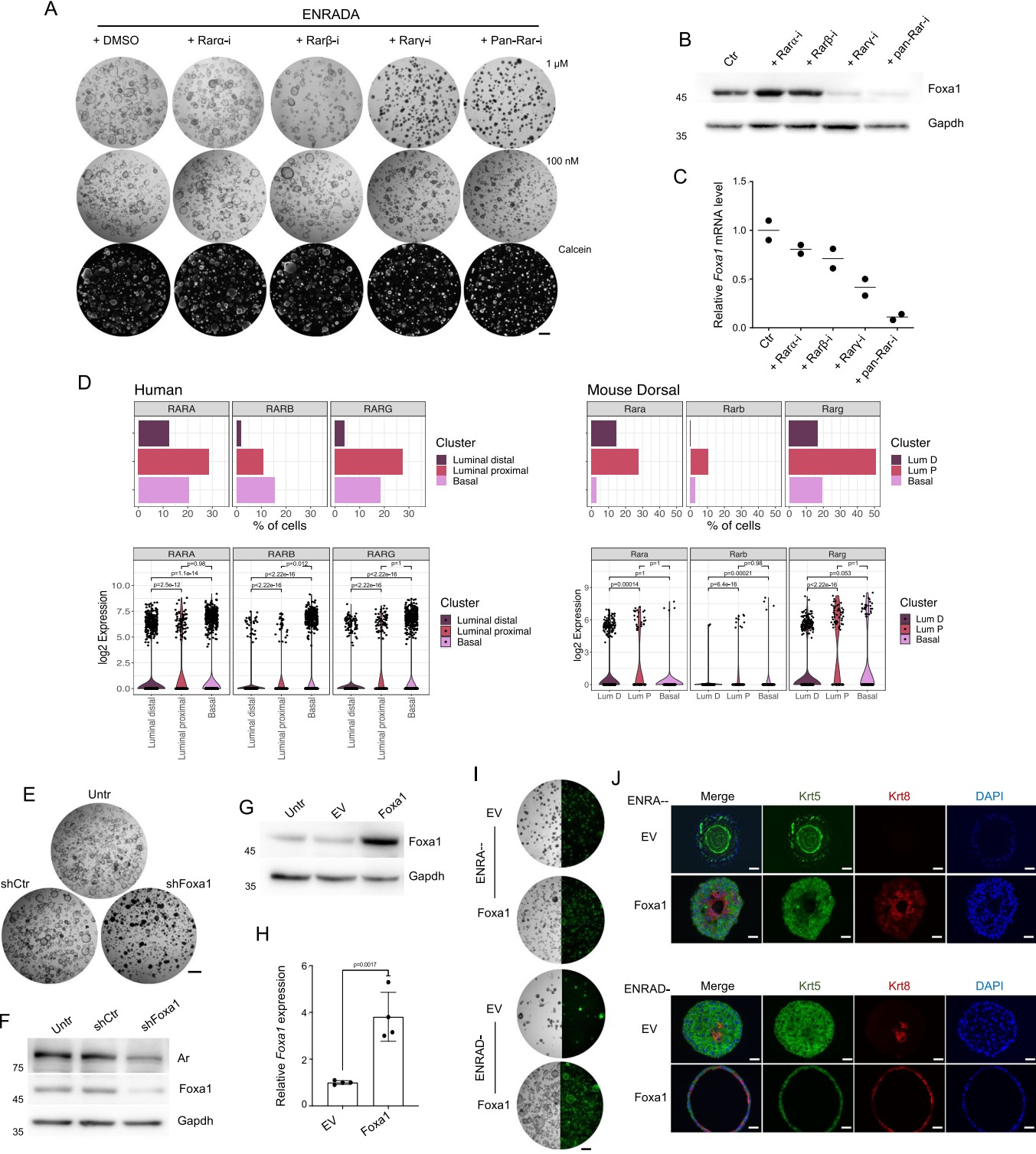

luminal lineage transcription factors (e.g., *Foxa1*) (Toivanen and Shen, 2017), genes involved in epithelial barrier establishment (e.g., *Cldn4*) (Garcia et al, 2018) and luminal cells function (e.g., *Tmprss2, Krt7, Krt8*) (Cambuli et al, 2022; Crowley et al, 2020; Karthaus et al, 2020) among the principal targets of Foxa1 transcriptional activity (Figs. 4B–I and EV4A,B and Dataset EV4).

In addition to the primary activity of Foxa1 on crucial epithelial genes, our analysis revealed widespread Foxa1-mediated reprogramming of Ar. In mPrOs[EV], Ar occupied over 3000 genomic sites, including ~2500 putative enhancers and >500 gene promoters (Figs. 4B–F and EV4C–H, and Dataset EV2). Foxa1 expression in mPrOs[Foxa1] was associated with the extensive reprogramming of the

◄  **Figure 3.  A Rarγ-Foxa1 transcriptional cascade is essential for the retinoic acid control of glandular identity in adult prostate progenitors.**

(A) mPrOs morphology upon administration of RARs inhibitors at different concentrations (upper and middle panels). Calcein staining determines mPrOs viability (lower panels). Scale bars, 1 mm. $N = 3$ independent biological replicates. (B) Representative Western blot analysis of Foxa1 expression in mPrOs treated with the different RAR inhibitors for 6 days. Gapdh is used as loading control. $N = 3$ independent biological replicates. (C) RT-qPCR analysis of Foxa1 expression in mPrOs treated with the RARs inhibitors. Data are presented as mean value ± s.d. of $n = 2$ independent biological replicates (Ctr, mean = 1; Rarα-i, mean = 0.8; Rarβ-i, mean = 0.71; Rarγ-i, mean = 0.41; pan-Rar-i, mean = 0.11). (D) Percentage of cells (bar plots) and expression levels (violin plots) of, *RARα RARβ*, and *RARγ* genes in epithelial cell populations of human and mouse normal prostate (Data ref: Crowley et al, 2020; Appendix Fig. S2). The *p*-values were calculated with the Mann–Whitney U Test. (E, F) Phenotypic response of mPrOs cultured in ENRADA to Foxa1 knock-down. Scale bar, 1 mm (E). Western blot showing reduction of Foxa1 and Ar level in mPrOs stably transduced with shRNAs against Foxa1 (F). Untransduced mPrOs and mPrOs expressing not targeting shRNAs (shCtr) are used as controls. Gapdh is used as loading control. $N = 3$ independent biological replicates. (G, H) Western blot analysis of Foxa1 expression in mPrOs grown without DHT and ATRA (ENRA--) untransduced (Untr), stably transduced with an empty vector (EV) or with a vector expressing mouse Foxa1 (Foxa1). Gapdh is used as loading control (G). RT-qPCR analysis of Foxa1 RNA expression in EV and Foxa1 mPrOs cultured in ENRA-- medium (H). Data are presented as mean value ± s.d. of $n = 4$ independent biological replicates. Unpaired t-test, $p < 0.05$ was considered statistically significant. (I, J) Morphological comparison of wild-type and transduced (EV and Foxa1) mPrOs cultured without ATRA and with or without DHT (ENRAD-, ENRA--) (I). Scale bar, 1 mm. $N > 3$ independent biological replicates. Immunofluorescence analysis of Krt5 (basal) and Krt8 (luminal) markers in the different conditions. Nuclei are stained with DAPI (J). Scale bar 50 μm. Source data are available online for this figure.

Ar cistrome. In mPrOs$^{Foxa1}$, Foxa1 was found in place of Ar at ~35% of the putative enhancers (distal elements, DE; Ar-bound (A) DE in mPrOs$^{EV}$ (0)/A0 $n = 2442$; A0 where Foxa1 (F) was in place $(p)$ of Ar in mPrOs$^{Foxa1}$ (1), F1$_a$ $(p)$ $n = 761$) and at ~20% of the promoters (Promoter elements, PE; A0 $n = 562$, F1$_a$ $(p)$ $n = 110$) bound exclusively by Ar in mPrOs$^{EV}$. Finally, ~70% of the genomic loci occupied by both transcription factors in mPrOs$^{EV}$ (DE + PE; FA0 $n = 250 + 31$) were exclusively occupied by Foxa1 in mPrOs$^{Foxa1}$ (DE + PE; F1$_b$ $n = 179 + 23$) (Fig. 4C–F). In the group of loci occupied by Ar in mPrOs$^{EV}$ and in which Foxa1 was found in place of Ar in mPrOs$^{Foxa1}$ (F1$_a$ $(p)$), we found the distal elements (DE) of the progenitor markers *Wfdc2*, *Krt19*, *Sox5*, *Fgf1* and *Runx2* and the luminal-associated genes *Krt8*, *Cldn4* and *Steap4* (Cambuli et al, 2022; Crowley et al, 2020; Karthaus et al, 2020; Henry et al, 2018; Mevel et al, 2020; Steiner et al, 2023), which result upregulated upon ATRA treatment (Figs. 4C–F and EV4C, and Dataset EV4). Additional critical DEGs, such as the key progenitor marker *Clu* (Crowley et al, 2020; Karthaus et al, 2020) and *Il33*, a cytokine involved in epigenetic reprogramming in epithelia (Alonso-Curbelo et al, 2021), displayed Foxa1 in place of Ar (F1$_a$ $(p)$) at genes' promoter elements (PE) (Figs. 4C–F and EV4D, and Dataset EV4). Unexpectedly, the intersection of ChIP-seq (A1 + A1$_a$ + A1$_b$ and F1$_a$ $(a)$ + FA1, PE and DE) and RNA-seq analyses did not yield clear insights into the role of Ar in epithelial differentiation and lumenogenesis (Figs. 4C–F and EV4E–H, and Dataset EV4).

## The hotspot Foxa1$^{F254E255}$ prostate cancer mutant is impaired in promoting luminal identity in prostate progenitors

*FOXA1* is altered in ~12% of prostate cancer patients, predominantly through single-residue variants and short indels (~8.5% of cases). Among these mutations, about half (~4.25%) occur within the Wing2 region (between H247 and E269) of the Forkhead DNA-binding domain (FKHD), which can be considered a mutational hotspot (Adams et al, 2019). A previous systematic phenotypic and molecular analysis of FOXA1 mutants in prostate organoids concluded that Wing2 alterations are gain-of-function variants conferring an enhanced pro-luminal differentiation program (Adams et al, 2019). Under standard organoid culture conditions, limited RA signaling results in low levels of Foxa1 and only a small fraction of prostate progenitors acquires luminal identity. We thus set out to investigate the role of the most common FOXA1

mutations in our optimized prostate organoid model, which is characterized by enhanced luminal differentiation. In-frame indels are a common type of cancer mutations affecting *FOXA1* (Adams et al, 2019; Arruabarrena-Aristorena et al, 2020), and F254E255 is one of the most frequent variants affecting prostate cancer patients (Fig. 5A).

We generated mouse prostate progenitor lines stably expressing F254E255 or two FOXA1 missense mutations occurring in the FKHD domain either before (D226N) or within the Wing2 region (H247Q). Organoids transduced with the empty vector (mPrOs$^{EV}$) served as control (Figs. 5B and EV5A). All three Foxa1 mutants were expressed at similar levels than exogenous Foxa1 wild type (Foxa1$^{wt}$), displaying a three- to four-fold increase in comparison to untransduced and mPrOs$^{EV}$ control organoids (Fig. 5C,D). In the absence of DHT and RA signaling (ENRA-- conditions with B27 zero), control organoids (mPrOs$^{EV}$) rarely showed well-shaped hollow organoids (Fig. 5B,E). As expected, the frequency of well-shaped hollow organoids increased in those expressing wild-type Foxa1 (mPrOs$^{WT}$), while the three *Foxa1* mutant organoid lines mainly generated compact spheroids, similar to controls (mPrOs$^{EV}$) (Fig. 5B,E). DHT administration (ENRAD- conditions) rescued the ability of mPrOS$^{WT}$, mPrOS$^{D226N}$ and mPrOs$^{H247Q}$ to form hollow organoids, whereas the frequency remained < 5% in mPrOs$^{F254E255}$ (Fig. 5E–G). Immunofluorescence and Western blotting experiments for prostate basal (Krt5) and luminal (Krt8) cell markers suggested a slight decrease in the ability of Foxa1 mutants D226N and H247Q to promote differentiation and expansion of luminal progenitors, a phenotype that was more severe for the F254E255 mutant (Fig. 5G,H). Despite unperturbed nuclear localization of wild-type and mutants Foxa1 (Figs. 5I and EV5B), reanalysis of the publicly available Foxa1 and Ar ChIP datasets in prostate organoids (Adams et al, 2019) revealed low occupancy of Foxa1$^{F254E255}$ at distal and proximal DNA elements (Figs. 5J,K and EV5C). Peak numbers of Foxa1$^{F254E255}$ were markedly distinct from wild-type Foxa1 and comparable to the EV control, as was its ability to displace Ar from DE and PE genome-wide (Figs. 5J,K and EV5C).

## Molecular modeling of the FOXA1$^{F254E255}$ FKHD domain is consistent with impaired DNA binding ability

To gain a structural understanding of the impact of FOXA1 FKHD domain variants on DNA binding we computationally modeled their interactions (Fig. 6). Accelerated Molecular dynamics

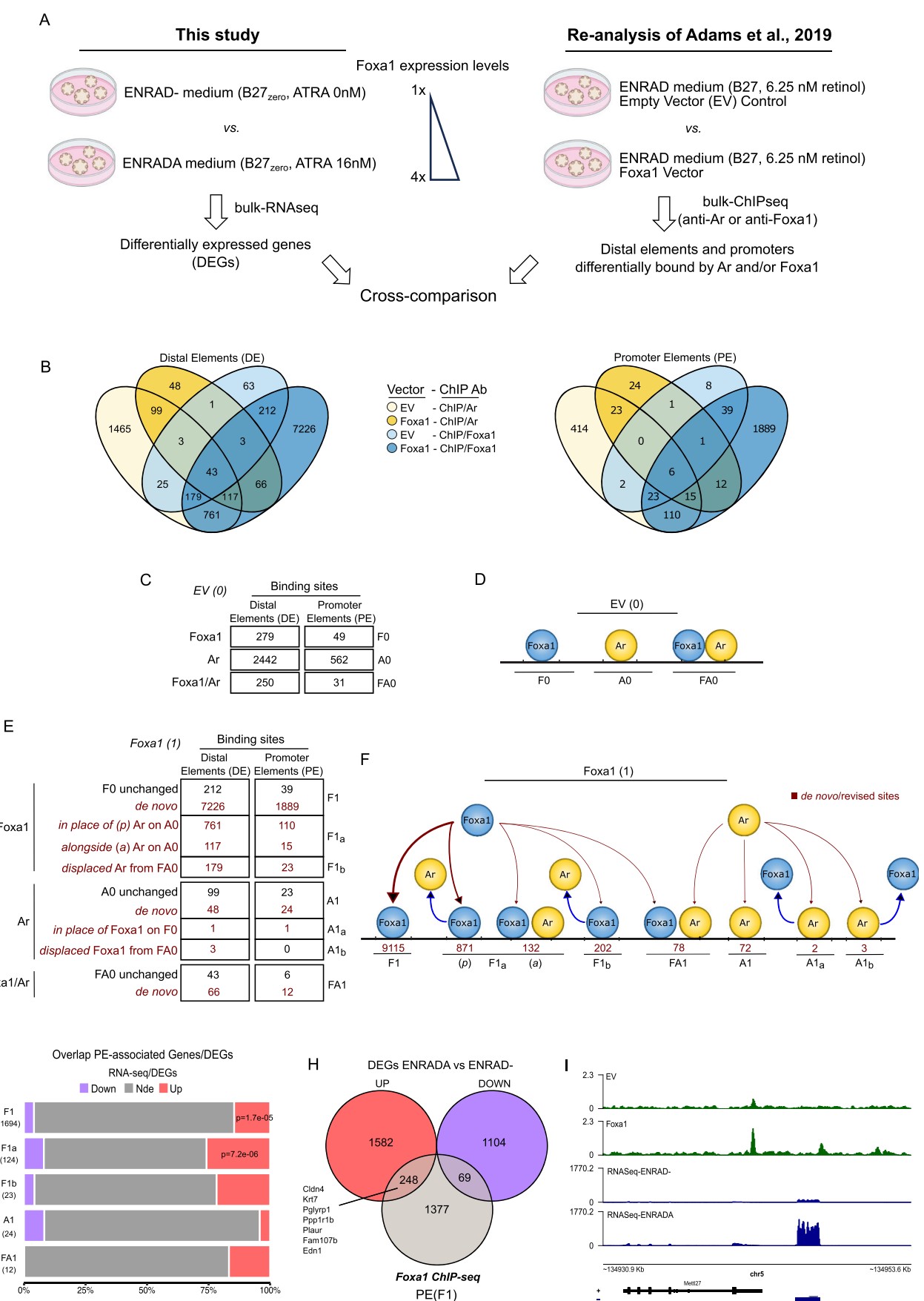

**Figure 4. Foxa1 occupies distal and promoter elements of key luminal progenitor genes and reshape androgen receptor binding genome-wide.**

(A) Schematic representation of the cross-comparison study of RNA-seq analysis performed in this study and published ChIP-seq datasets (Adams et al, 2019). (B) Venn diagram showing the binding sites of Ar and Foxa1 in the genome of mPrOs expressing endogenous (empty vector, EV) or exogenous (Foxa1) Foxa1. Distal elements (±2.5 kb away from an annotated gene promoter) are displayed on the left, whereas promoter sites are shown on the right. ChIP-seq data are from Adams et al, 2019 ($n = 2$ replicates per condition). (C–F) Numerical (C and E) and graphical (D and F) representation of Foxa1 and Ar cistromes at both distal (DE) and promoter (PE) elements with endogenous levels of Foxa1 (EV (0)) and upon its overexpression (Foxa1 (1)). The number of de novo and pre-existing but rearranged PE/DE sites are indicated in red. (G) Percentage overlap of ChIP-seq promoter elements (PE)-associated Genes (Adams et al, 2019) with Upregulated and Downregulated DEGs from RNA-seq (comparison ENRADA vs ENRAD-, this work). The significance of the overlap has been determined by a hypergeometric test. Each bar represents a set of genes (e.g., $n = 1694$) associated to a specific class of PE (e.g., de novo Foxa1-bound PE, F1). (H) Venn diagrams showing the overlap of differentially expressed genes in mPrOs cultured in ENRADA versus ENRAD, and exogenous Foxa1-bound promoter elements (PE (F1)) in the genome. Relevant upregulated genes in the intersection are highlighted. (I) Genomic snapshot of ChIP-seq ($n = 2$ pooled replicates) and RNA-seq ($n = 3$ pooled replicates) signals over the selected gene *Cldn4*.

simulations (aMD) of wild-type FOXA1 bound to DNA yielded a stable complex, with most sampled conformations remaining around ~3 to 4 Å from the original crystallographic structure for both interacting macromolecules (e.g., FOXA1 and the double-stranded DNA molecule) (Fig. 6A, left panel-black line). The double-stranded DNA within the FOXA1$^{F254E255}$-DNA complex was also stable with sampled conformations within ~4 Å of the reference structure (Fig. 6A, left panel-red line). However, compared to wild-type FOXA1 (Fig. 6A, right panel-black line), mutant FOXA1$^{F254E255}$ deviated significantly from its initial conformation, averaging ~7 Å (Fig. 6A, right panel-red line). The F254 and E255 residues deleted in mutant FOXA1$^{F254E255}$ are part of an α-helix at the C-terminus of the FKHD. The α-helix remained very stable during the simulation of wild-type FOXA1 binding to DNA (Fig. 6B,D (black line)), whereas it completely unfolded in mutant FOXA1$^{F254E255}$ (Fig. 6C,D (red line)). Residue F254 is buried in a cavity formed by residues F72, I20, L90, C71, T24 and Y103, while the side chain of E255 is involved in hydrogen bond interactions with K189 and G184 (Fig. 6B). These interactions were well-maintained during the MD simulation with wild-type FOXA1, whereas they were completely lost in FOXA1$^{F254E255}$, leading to unfolding of the α-helix at the C-terminus of the FKHD domain resulting in high flexibility. These structural changes in mutant FOXA1$^{F254E255}$ are consistent with a reduction in the number of FOXA1-DNA contacts, causing loss of DNA affinity (Fig. 6E).

## Discussion

The ability of RA signaling to promote cell lineage commitment via FOXA1 (*alias* HNF-3alpha) was previously reported in the context of embryonal development (e.g., neuronal tissue, endoderm) (Jacob et al, 1994, 1999; Tan et al, 2010; Taube et al, 2010). Recently, the vitamin A metabolite Retinoic Acid (RA) has been shown to restrict the lineage plasticity of adult stem cells of the skin (Tierney et al, 2024), shedding light on a pivotal role of RA signaling in adult progenitor lineage commitment, and its deregulation during tumorigenesis.

Our work on mouse prostate organoids specifically identifies *Foxa1* as a crucial target of RA-RARγ signaling in the prostate. In rodent development, *Foxa1* is expressed in the entire urogenital epithelium (UGE) before prostate induction, while it is restricted to the luminal compartment thereafter. *Foxa1* genetic ablation causes loss of luminal secretory cells, prostatic hyperplasia and transdifferentiation, in line with an instructing role of Foxa1 in luminal lineage commitment (Gao et al, 2005; DeGraff et al, 2014). Consistent with these findings, we have shown that nanomolar

amounts of all-trans retinoic acid (ATRA) induce the expression of mouse prostate luminal progenitor genes (*Krt4, Krt7, Wfdc2, Clu, Ppp1r1b*) as well as genes (*Tjp1, Tjp3, Ocln, Cldn4, Cldn7*) encoding tight- and gap-junction proteins, all related to programming or establishing a functional luminal compartment. Mechanistically, the pioneering activity of Foxa1 in the prostate epithelium has been attributed to its ability to modulate androgen signaling by cooperating with Ar at promoter and enhancer regions of specific gene subsets (Adams et al, 2019; Cirillo et al, 2002; Gao et al, 2003; Pomerantz et al, 2015). Yet, we found that Foxa1 binds to the distal and/or promoter regions of many genes independently of Ar, indicating a direct and pivotal role enforcing the luminal progenitor fate. Still, the androgen pathway is required for proper lumenogenesis, as shown by the formation of well-shaped hollow organoids.

Although this work advances the ability to model the prostate luminal progenitor compartment in vitro, prostate organoid systems are still limited in the ability to efficiently generate fully differentiated luminal cells, partially restricting the physiological relevance of this model. We expect that continuous progress modeling prostatic functions in vitro will further extend our comprehension of the underlying molecular mechanisms, including the interplay between FOXA1 and AR transcriptional regulation. Notably, while complete withdrawal of RA signaling from the growth medium generates compact spheres of proliferating cells that are almost invariably positive for the basal marker Krt5, we did rarely observe a few ectopic Krt8 positive cells in the center of the Krt5+ mass of cells, pointing to the possible role of still unknown signaling mediators in the specification of luminal progenitors.

The availability of a prostate organoid model with enhanced luminal identity, and the identification of Foxa1 as a major target of RA signaling led us to explore its role in prostate cancer. Recently, recurrent missense mutations have been identified in human prostate and breast cancers as drivers of epithelial transformation and tumorigenesis (Adams et al, 2019; Arruabarrena-Aristorena et al, 2020). FOXA1 mutations have been generally hypothesized as enhancers of transcriptional activity on canonical and de novo target genes, and as causal agents of aberrant androgen and estrogen receptor functions (Adams et al, 2019; Arruabarrena-Aristorena et al, 2020). Notably, the F254 residue, which plays a crucial role in stabilizing the α-helix at the C-terminus of the FKHD domain of FOXA1, is deleted in a substantial fraction of prostate and breast cancer-associated indels (Fig. 4A). We have shown that Foxa1$^{F254E255}$ mutant fails to promote luminal identity in mouse prostate organoids due to reduced DNA binding stability and impaired transcriptional activity.

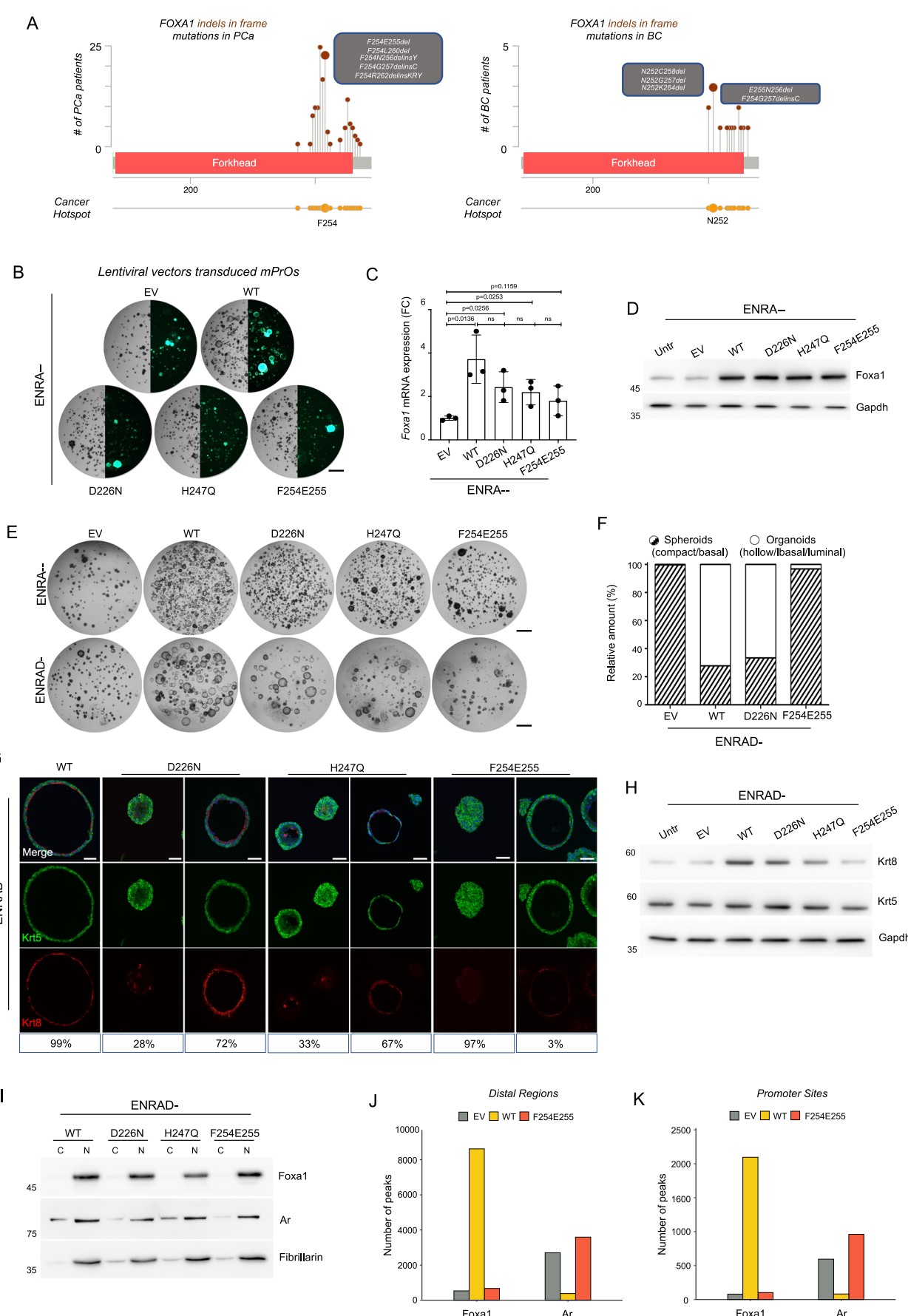

◄ **Figure 5. The hotspot Foxa1$^{F254E255}$ prostate cancer mutation is unable to promote luminal identity in prostate progenitors.**

(A) Indels mutations of the alpha-helix region at the C-terminal part of the Forkhead domain of FOXA1 identified in prostate (left) and breast (right) cancers (cBioportal/Cosmic databases). (B–D) Brightfield and fluorescence images of mPrOs grown without ATRA and DHT (ENRA--) expressing wild-type Foxa1 or its mutant forms D226N, H247Q, or F254E255. mPrOs transduced with the empty vector (EV) were used as controls (B). Scale bar, 1 mm. RT-qPCR analysis of Foxa1 RNA expression in transduced mPrOs cultured in ENRA-- medium (C). Western blot analysis of Foxa1 protein levels in transduced mPrOs (D). Gapdh is used as loading control. Data are presented as mean value ± s.d. of $n = 3$ independent biological replicates. Unpaired t-test, $p < 0.05$ was considered statistically significant. (E, F) Morphological analysis of mPrOs transduced with Foxa1 variants upon re-administration of DHT (E) and quantification of compact versus hollow organoids (F). Scale bars: 1 mm. $N = 2$ independent biological replicates. (G, H) Immunofluorescence (G) and Western blot (H) analyses of Krt5 (basal) and Krt8 (luminal) markers in mPrOs cultured in presence of DHT but not ATRA (ENRAD-) and expressing exogenous wild-type Foxa1 or its mutant forms. Gapdh is used as loading control. Scale bar, 50 μm. $N = 3$ independent biological replicates. (I) Biochemical fractionation of nuclear (N) and cytosolic (C) compartments showing nuclear localization of wild type and mutant form of Foxa1. Ar and Fibrillarin are used as nuclear markers and loading controls. (J, K) Number of peaks identified by ChIP-seq in distal regions (J) and gene promoters (K) for Foxa1 and Ar in mPrOs stably transduced with wild-type Foxa1 (WT), Foxa1$^{F254E255}$ (F254E255) or the empty vector (EV) and cultured with DHT but not ATRA, as reported in Adams et al, 2019. Source data are available online for this figure.

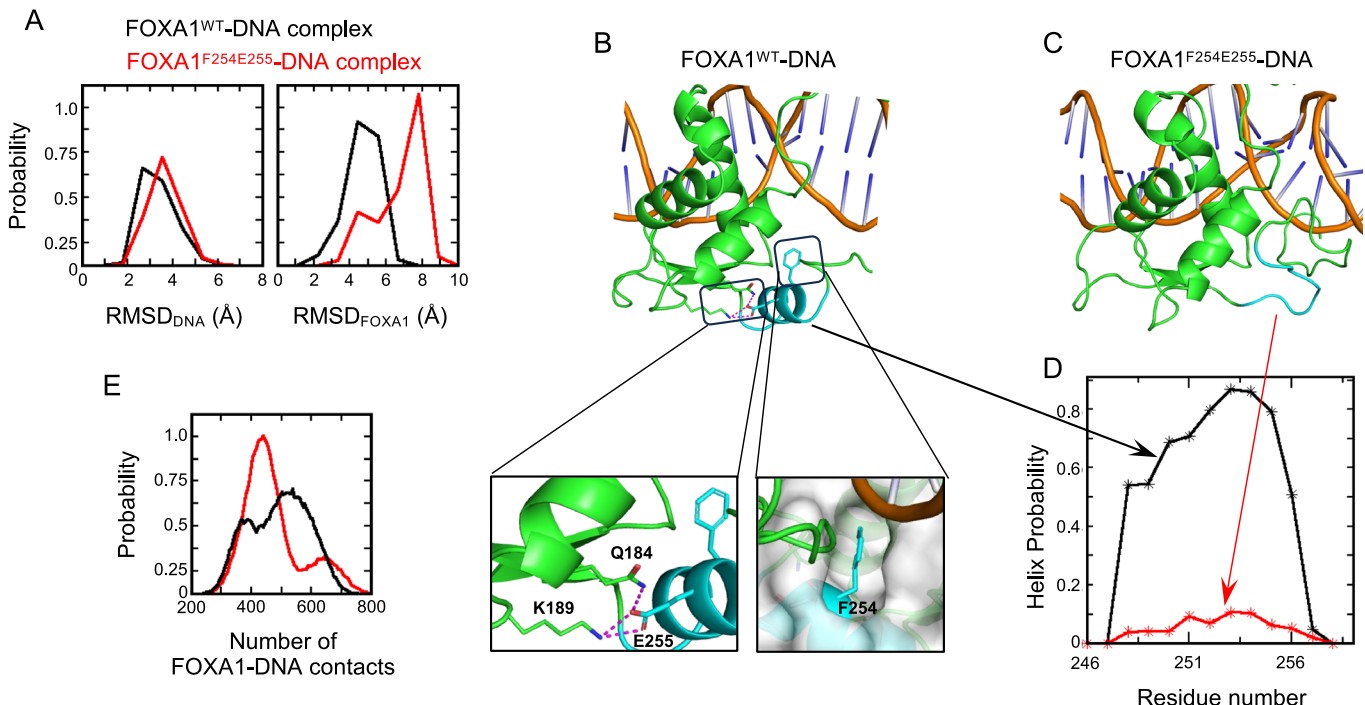

**Figure 6. Molecular modeling of the FOXA1$^{F254E255}$ FKHD domain is consistent with impaired DNA binding ability.**

(A) Distribution of Root Mean Squared Deviation (RMSD) of conformations of FOXA1$^{WT}$–DNA (black) and FOXA1$^{F254E255}$–DNA (red) sampled during aMD simulations. (B, C) Snapshot from MD simulation highlighting the C-terminal α-helix (cyan) of FKHD of the FOXA1$^{WT}$–DNA complex (B), which becomes disordered (cyan) in the FOXA1$^{F254E255}$–DNA complex (C). In the case of the FOXA1$^{WT}$–DNA complex (B), residues Q184, K189, and E255 are shown as sticks and interactions between them are highlighted in dashed lines (magenta). The residue F254 is shown as sticks and the region around it is shown as surface (gray) highlighting that the sidechain of F254 is buried in the cavity. (D) Distribution of the helix probability (from the MD simulations) of the conformation of the C-terminal α-helix of FKHD of FOXA1 sampled during FOXA1$^{WT}$–DNA (black) and FOXA1$^{F254E255}$–DNA (red) complexes. (E) Probability of the number of FOXA1–DNA contacts of FOXA1$^{WT}$ (black) and FOXA1$^{F254E255}$ (red) (from the MD simulations).

FOXA1 loss-of-function mutations could represent a genetic vulnerability in a subset of cancer patients. FOXA1 binds DNA as a monomer on A(A/T)TRTT(G/T)R(C/T)T(C/T) consensus elements, or as a homodimer on compact palindromic DNA elements (diverging half-sites-DIV) (Cirillo and Zaret, 2007; Wang et al, 2018). In tumor cells with loss-of-function mutations of FOXA1 that preserve the ability to form homodimers, concomitant induction of both wild-type and mutant alleles will presumably result in a dominant negative effect of the mutant protein on the regulation of DIV elements. In contrast, a benefit of FOXA1 overexpression should be expected on DIV controlled genes in the presence of loss-of-function mutant alleles unable to homodimerize, and on targeted genes where FOXA1 works as a monomer. Accordingly, preliminary experiments in mouse prostate organoids bearing the F254E255 mutation of Foxa1 showed the ability of ATRA to rescue the formation of large hollow organoids in vitro, likely caused by the induction of the endogenous Foxa1$^{WT}$ alleles.

Whether and how this specific impairment of FOXA1 transcriptional function may impact tumor prognosis and treatment is still unknown but it deserves close attention for its potential clinical relevance. ATRA and synthetic retinoids such as fenretinide (4-HPR) or etretinate (Tegison) have been clinically tested in

several solid tumors characterized by dysfunctional RA signaling (Costantini et al, 2020; Ozgun et al, 2021). To date, no clinical trial has demonstrated efficacy, and activation of the retinoic pathway remains a clinical option only for the treatment of PML-RARα Acute Promyelocytic Leukemia (APL). Noteworthy, subgroups of patients with superficial papillary or resected high-risk non-muscle invasive bladder cancer showed reduced recurrence rate and cancer progression (Sabichi et al, 2008; Studer et al, 1995; Alfthan et al, 1983), while few patients with advanced breast cancer achieved partial response or had stable disease (Sutton et al, 1997). In this scenario, orthotopic transplants in syngeneic wild type adult mice of mPrOs (Cambuli et al, 2022) carrying tumor-associated Foxa1 mutations will provide a valuable preclinical platform to test the efficacy of RA signaling in counteracting the tumorigenic process according to the specific class of Foxa1 mutations.

Overall, our study adds new important insights to the network of signaling pathways and molecular circuits regulating prostatic luminal lineage commitment and adult tissue homeostasis and paves the way for more accurate assessments of retinoid derivatives for the treatment of solid tumors that can be stratified by FOXA1 mutations.

# Methods

### Reagents and tools table

| Reagent/Resource | Reference or Source | Identifier or Catalog Number |
| --- | --- | --- |
| **Experimental Models** | | |
| Mouse Prostate Organoids | This manuscript | N/A |
| **Recombinant DNA** | | |
| pMSCV-Neo-GFP/Foxa1 | Addgene | #105506 |
| pMSCV-Neo-GFP | Addgene | #105505 |
| pMSCV-Neo-GFP/Foxa1$^{D226N}$ | This manuscript | N/A |
| pMSCV-Neo-GFP/Foxa$^{H247Q}$ | This manuscript | N/A |
| pMSCV-Neo-GFP/Foxa1$^{F254E255}$ | This manuscript | N/A |
| LEPG-shFoxa1 2959 | Cristopher Vakoc's Lab Roe et al, 2017 | N/A |
| LEPG-shRLuc | Cristopher Vakoc's Lab Roe et al, 2017 | N/A |
| **Antibodies** | | |
| Ar | Santa Cruz | sc-816 |
| Foxa1 | Abcam | ab55178 |
| Gapdh | ThermoFisher Scientific | MA515738 |
| β-actin | Merck | A2228 |
| Cytokeratin 5 | Abcam | ab905901 |
| Cytokeratin 8 | Abcam | ab53280 |
| Zo-1/Tjp1 | Life Tech | 339100 |
| Zo-3/Tjp3 | BioTechne | LS-C313103 |
| Claudin 4 | Life Tech | 329400 |
| Claudin 7 | Life Tech | 349100 |

| Reagent/Resource | Reference or Source | Identifier or Catalog Number |
| --- | --- | --- |
| E-cadherin | Cell Signaling | BK3195T |
| Anti-mouse HRP | Cell Signaling | 7076 |
| Anti-rabbit HRP | Cell Signaling | 7074 |
| Mouse Alexa Fluor 488 | LifeTech | A21202 |
| Chicken Alexa Fluor 633 | LifeTech | A21103 |
| Rabbit Alexa Fluor 594 | LifeTech | A21207 |
| **Oligonucleotides and other sequence-based reagents** | | |
| Foxa1 Fw 5′-CATGAGAGCAACGA CTGGAA-3′ | Integrated DNA Technologies | N/A |
| Foxa1 Rev 5′-TTGGCGTAGGACAT GTTGAA-3′ | Integrated DNA Technologies | N/A |
| Tbp Fw 5′-CGGTCGCGTCATTT TCTCCGC-3′ | Integrated DNA Technologies | N/A |
| Tbp Rev 5′-GTGGGGAGGCCA AGCCCTGA-3′ | Integrated DNA Technologies | N/A |
| Gapdh Fw 5′-GAGAGTGTTTCCT CGTCCCG-3′ | Integrated DNA Technologies | N/A |
| Gapdh Rev 5′-ACTGTGCCGTTGA ATTTGCC-3′ | Integrated DNA Technologies | N/A |
| FOXA1 mutated cDNAs | Twist Biosciences | N/A |
| **Chemicals, Enzymes and other reagents** | | |
| Egf | PeproTech | 315-09 |
| Noggin | PeproTech | 120-10C |
| A83-01 | Tocris | 2393 |
| DHT | Merck | 10300 |
| ATRA | Merck | R2625 |
| Y-27632 | Merck | Y0503 |
| RARα antagonist (BMS195614) | Cayman | 16029.1 |
| RARβ antagonist (LE135) | Cayman | 14415.1 |
| RARγ antagonist (LY2955303) | Cayman | 25833.1 |
| Pan-RAR antagonist (AGN 193109) | Cayman | 23975.5 |
| Enzalutamide (MDV-3100) | Vinci Biochem | CAY-11596-5 |
| **Software** | | |
| Bowtie (v1.3) | Langmead et al, 2009 | N/A |
| MACS2 v2.2.7 | Zhang et al, 2008 | N/A |
| Irreproducible Discovery Rate (IDR v2.0.3) | Li et al, 2011 | N/A |
| ChIPseeker v1.28.3 | Yu et al, 2015 | N/A |
| ChIPpeakAnno v3.32 | Zhu et al, 2010 | N/A |
| deepTools v3.5.1 | Ramírez et al, 2014 | N/A |
| ComplexHeatmap | Gu et al, 2016 | N/A |
| STAR-v2.6.0 | Dobin et al, 2013 | N/A |
| Trimmomatic-v0.35 | Bolger et al, 2014 | N/A |

| Reagent/Resource | Reference or Source | Identifier or Catalog Number |
|---|---|---|
| HTSeq-count v0.5.4 | Anders et al, 2015 | N/A |
| DEseq2 | Love et al, 2014 | N/A |
| Metascape | Zhou et al, 2019 | N/A |
| Amber ff14SB | Maier et al., 2015 | N/A |
| FF99BSC0 | Pérez et al, 2007 | N/A |
| aMD | Pierce et al, 2012; Hamelberg et al, 2007 | N/A |
| VMD | Humphrey et al, 1996 | N/A |
| Pymol | PYMOL | N/A |
| Prism | GraphPad | N/A |
| Alliance LD2 | UVITEC | N/A |
| BioRender | BioRender | N/A |
| Inkscape | Inkscape | N/A |

## Materials availability

All unique/stable reagents generated in this study are available from the Lead Contact upon reasonable request or with a completed Material Transfer Agreement.

## Mouse housing and husbandry

Housing systems followed FELASA guidelines and recommendations concerning animal welfare, health monitoring and veterinary care, in compliance with the Directive 2010/63/UE and its Italian transposition D. L.vo 26/2014. Mice were monitored daily for general health and well-being and sentinel mice are used for quarterly monitoring for specific pathogens. Wild-type C57BL/6J (JAX #000664) mice were purchased from the Jackson Laboratory, wild-type CD-1 (CRL #022) mice were purchased from the Charles River Laboratories. Mice were housed in room with 21 °C temperature with 12 h light/dark cycle with light gradually rising at 7:00 a.m. and gradually decreasing at 7.00 p.m. A maximum of 5 animals were accommodated in IVC cages with food and water ad libitum and nesting materials and cardboard tunnels were provided as enrichment. Animals were sacrificed according to the European Communities Council Directive (2010/63/EU) and following the protocol approved by the Italian Ministry of Health and the University of Trento Animal Welfare Committee (642/2017-PR).

## Mouse prostate organoid cultures

Mouse prostate organoids were generated from prostate glands collected from adult (6-month-old) inbred C57BL/6J or outbred CD1 wild type males as described in (Cambuli et al, 2022). The following media were used (see Reagents and Tools Tables for small molecules used for organoid culture media):

ENRADA: AdDMEM 4+, 50 ng/ml Egf, 100 ng/ml R-Spondin1 conditioned medium (Cell Tech Facility at CIBIO, using Cultrex® Rspo1 cells following the guidelines from

Trevigen), 0.2 μM A83-01, 10 nM DHT, 16 nM ATRA. Stored in the dark at 4 °C for up to 1 week.

ENRAD-: AdDMEM 4+, 50 ng/ml Egf, 100 ng/ml Noggin, 10% R-Spondin1 conditioned medium (Cell Tech Facility at CIBIO, using Cultrex® Rspo1 cells following the guidelines from Trevigen), 0.2 μM A83-01, 10 nM DHT. Stored in the dark at 4 °C for up to 1 week.

ENRA--: AdDMEM 4+, 50 ng/ml Egf, 100 ng/ml Noggin, 10% R-Spondin1 conditioned medium (Cell Tech Facility at CIBIO, using Cultrex® Rspo1 cells following the guidelines from Trevigen), 0.2 μM A83-01. Stored at 4 °C for up to 1 week.

## Viral transduction

Organoids were dissociated to single cells, and ~50,000 cells used for viral transduction. Spinoculation was performed in a low-adhesion 96-well plate using 0.6–1.0 RTU/ml of lentiviral solution, supplemented with polybrene (4 μg/mL; Sigma-Aldrich, H9268) and ENRADA complete medium (Egf (50 ng/mL; PeproTech, 315-09), Noggin (100 ng/mL; PeproTech, 120-10C), R-Spondin1 (10% conditioned medium), A83-01 (200 nM; Tocris, 2393), dihydrotestosterone (10 nM; Merck, 10300), and ATRA (16 nM; Merck, R2625), or ENRA-- medium (without dihydrotestosterone and ATRA) supplemented with Y-27632 (10 μM; Calbiochem, 146986-50-7) to a final volume of 300 μL. The plate was centrifuged for 1 h at $600 \times g$, cells resuspended in 200 μL of ENRADA or ENRA-- medium supplemented with Y-27632 (10 μM) and incubated in suspension at 37 °C for 4–6 h. Then cells were mildly centrifuged ($300 \times g$, 5 min), cell pellet resuspended in 80% growth factor-reduced basement matrix (either Matrigel®, Corning, 356231; or BME-2®, AMSBIO, 3533) and seeded at the concentration of ~50,000 cells/mL by depositing at least six 40 μL drops at the bottom of a non-tissue culture treated plate. Domes were left to solidify for 15 min and covered with ENRADA or ENRA-- medium. Antibiotic selection started 2 days post-transduction. After approximately 2 weeks of antibiotic selection, transduced organoids expressed constitutively the green fluorescent protein (GFP). The following plasmids were used: pMSCV-Neo-GFP/FOXA1 (Addgene #105506) plasmid and the negative control pMSCV-Neo-GFP/Empty (Addgene #105505) were purchased on Addgene. Foxa1 mutated cDNAs (Twist Biosciences) were subcloned into pMSCV-Neo-GFP/Foxa1 after the enzymatic removal of the wild-type Foxa1 cDNA to generate pMSCV-Neo-GFP/Foxa1[D226N], pMSCV-Neo-GFP/Foxa1[H247Q], and pMSCV-Neo-GFP/Foxa1[F254E255]. LEPG-shFoxa1 2959 and LEPG-shRLuc were kindly provided by Cristopher Vakoc's Lab (Roe et al, 2017).

## Immunofluorescence studies

Organoids were cultured for 5–7 days, released from the basement membrane using a recovery solution—including Dispase II (1 mg/mL)—seeded in a neutralized collagen type-I solution (Corning, 354249) and cultured for additional 24 h before fixing them with 4% paraformaldehyde (Sigma-Aldrich, P6148) for 5 h, at room temperature. Prostate tissue was harvested and immediately fixed using the same conditions. Paraffin embedding and 5 μm sectioning were carried out according to standard procedures. For immuno-localization studies, antigen retrieval was performed with citrate-based buffer (pH 6.0) (Vector Lab, H3300) in a microwave

(90–100 °C) for 20 min. Slides were incubated in blocking solution (5% FBS + 0.1% Triton-X in PBS) for 1 h at room temperature, and with primary antibodies at 4 °C overnight. Spectrally distinct fluorochrome-conjugated antibodies were incubated for 2 h at room temperature. Slides were counterstained with Hoechst 33342 (Abcam, ab145597), and FluorSave mounting medium (Merck, 345789) applied before the coverslip. Mouse prostate was isolated, fixed in 4% paraformaldehyde for 20 min at room temperature and processed for immunolocalization studies as described for organoids. Primary and secondary antibodies used in this study are listed in the Reagents and Tools Tables.

## RNA extraction

Total RNA was extracted using the RNeasy Plus Micro kit (Qiagen, 74034) according to the manufacturer instructions, and analyzed with an Agilent BioAnalyzer 2100 to confirm integrity (RIN > 8), before proceeding with downstream applications.

## Semi-quantitative and quantitative PCR

RNA was retrotranscribed into cDNA using the iScript™ cDNA synthesis kit (BioRad, 1708891). PCR was performed using Phusion Universal qPCR Kit (Life Tech, F566L). PCR products were loaded in a 2% agarose gels, supplemented with Atlas DNA stain and separated by standard gel electrophoresis. DNA gels images were acquired with an UV scanner (UVITEC). Real-time quantitative PCR was performed with qPCRBIO SyGreen Mix (PCRBiosystems, PB20.14-05), according to the manufacturer instructions, and the CFX96 Real Time PCR thermocycler (Bio-Rad). The following primers were used to evaluate the expression of *Foxa1*, Fw: 5′-CATGAGAGCAACGACTGGAA-3′ and Rev: 5′-TTGGCGTAGGACATGTTGAA-3′; *Tbp*, Fw: 5′-CGGTCGCGTCATTTTCTCCGC-3′ and Rev: 5′-GTGGGGAGGC-CAAGCCCTGA-3′; *Gapdh*: Fw: 5′-GAGAGTGTTTCCTCGTCCCG-3′ and Rev: 5′-ACTGTGCCGTTGAATTTGCC-3′.

## ChIP sequencing data analysis

Published ChIP-seq fastq files were retrieved from GEO (Data ref: GSE128867). Reads were aligned to the mm10 reference genome with bowtie (v1.3) and peak calling was performed with MACS2 v2.2.7 in narrow mode, with parameters "--keep-dup all -m 3 30 --format BAMPE --pvalue 0.05". Due to the extremely low number of sequenced reads, we discarded two samples (R219S-FOXA1-rep1 and R219S-AR-rep1) from the dataset. Next, Irreproducible Discovery Rate (IDR v2.0.3) framework was used to select high reproducible peaks between the two replicate samples per condition (except for R219S). Only peaks with an IDR < 0.05 were kept. These remaining regions have been annotated with the R package ChIPseeker v1.28.3 (Yu et al, 2015), with a range to define a promoter peak of ± 2.5 kb. Distal elements were defined as all the non-promoter regions, including Distal Intergenic, UTR, Intronic, and Exonic. Finally, the peaks overlap between different condition was performed with the R package ChIPpeakAnno v3.32 (Zhu et al, 2010). bamCoverage from deepTools v3.5.1 (Ramírez et al, 2014) was used to create BigWig files, with parameters "--binSize 50 --extendReads". BigWigs from the two replicates were merged with wiggletools mean (Zerbino et al, 2014). deepTools functions

computeMatrix and plotHeatmap were used to visualize the ChIP-seq signals over the called peaks.

## ChIP-seq and RNA-seq data integration

To allow the comparison between gene expression and transcription factor binding data, we selected differentially expressed genes in ENRADA vs ENRAD- from the bulk RNA-seq analysis (| log2FC| ≥ 1 and adj. *P*-value < 0.05). Both up- and down-regulated genes were intersected with the target genes from the ChIP-seq. Specifically, each distal or promoter peak was assigned to its nearest gene based on the distance from the promoter region, ensuring a single gene (target) association for each peak. Venn diagrams and heatmaps were created to visualize the overlap. Heatmaps were plotted with the R package ComplexHeatmap (Gu et al, 2016). Moreover, the hyper-geometric test was used to calculate the significance of the intersection between upregulated genes and transcription factor targets from the ChIP-seq. As background for the test, it was set the total number of expressed genes from the RNA-seq (normalized read count across samples ≥10).

## RNA sequencing data analysis

cDNA libraries were prepared with TruSeq stranded mRNA library prep Kit (Illumina, RS-122-2101) using 1 µg of total RNA. RNA sequencing was performed on an Illumina HiSeq 2500 Sequencer using standard Rapid Run conditions at the Next-Generation Sequence Facility of University of Trento (Italy). The reads obtained from each sample were on average 25 million, 100 base pairs long, and single-ended. Adapter trimming and quality-base trimming were performed on the FASTQ file generated by the Illumina HiSeq2500 sequencing machine using Trimmomatic-v0.35 (Bolger et al, 2014). The reads were aligned to the Mus Musculus genome (mm10) using STAR-v2.6.0 (Dobin et al, 2013) with a maximum mismatch of two and default settings for all other parameters. Then, uniquely mapped reads were selected, and individual sample reads were quantified using HTSeq-count v0.5.4 (Anders et al, 2015) tool to obtain gene-level raw counts based on GRCm38.92 Ensembl (www.ensembl.org) annotation. Individual sample counts were normalized via relative log expression (RLE) using DEseq2 (Love et al, 2014), which was also used to perform differential expression analyses. *P*-values were adjusted for multiple hypothesis testing using the method of Benjamini and Hochberg. Differentially expressed genes between each comparison were those genes with absolute fold-change >1 and adjusted *p*-value < 0.05. Heatmaps were created with the R package ComplexHeatmap (Gu et al, 2016). Specifically, the genes were annotated as indicated in Data ref: Crowley et al (2020) and hierarchical clustering was performed on the rows with the complete linkage method. The normalized counts obtained from DESeq2 are shown for each gene and condition. Functional enrichment analysis of rescued genes was performed using Metascape (Zhou et al, 2019), using GO biological processes gene sets. All genes in the genome were used as enrichment background. Terms with *p*-value < 0.01, minimum count of 3, and an enrichment factor >1.5 were collected and grouped into clusters based on membership similarities. The most statistically significant term within each cluster was chosen to represent the cluster. BigWig files have been generated with

deepTools v3.5.1 (Ramírez et al, 2014) with parameters "--binSize 50 --normalizeUsing RPKM". The three replicates per condition were merged with wiggletools mean (Zerbino et al, 2014).

## Single cell expression analysis

Log-2 transformed gene counts were downloaded from the Single-cell atlas of the mouse and human prostate (Data ref: Crowley et al, 2020). These data are available on the Broad Institute Single Cell Portal (https://singlecell.broadinstitute.org/single_cell/study/SCP1080/, SCP1081, SCP1082, SCP1083, SCP1084). The clusters' annotations for each cell type were retrieved as assigned by Crowley et al (2020) and used for the bar plot and violin plots generation. Wilcoxon rank test was used for the box plot comparisons.

## Subcellular fractionation and western blotting

Organoid cell pellets were lysed in RIPA buffer (50 mM Tris-HCl, pH 7.5, 150 mM NaCl, 1% Triton X-100, 1% sodium deoxycholate, 1% NP-40) supplemented with protease (Halt™ protease inhibitor cocktail, Life Tech, 87786) and phosphatase inhibitors (Phosphatase-Inhibitor Mix II solution, Serva, 3905501). NE-PER Nuclear and Cytoplasmic Extraction Kit (Life Tech, 78833) was used for nuclear/cytoplasmic fractionation according to the manufacturer instructions. Protein concentrations were measured using the BCA Protein Assay Kit (Pierce™ BCA Protein Assay kit, Thermo Fisher Scientific, 23225) and a Tecan Infinite M200 Plate Reader. Proteins were resolved via SDS-PAGE and transferred to polyvinylidene difluoride (PVDF) membrane (Merck, GE10600023) with a wet electroblotting system (Bio-Rad). The membranes were blocked with 5% non-fat dry milk or 5% BSA in TBS-T (50 mM Tris-HCl, pH 7.5, 150 mM NaCl, 0.1% Tween20) for 1 h at room temperature, then incubated with designated primary antibodies overnight at 4 °C. After washing, membranes were incubated with HRP-conjugated secondary antibody for 1 h at room temperature. ECL LiteAblot plus kit A + B (Euroclone, GEHRPN2235) was used to detect immunoreactive bands with an Alliance LD2 device and software (UVITEC). Primary and secondary antibodies used in this study are provided in the Reagents and Tools Tables.

## Molecular dynamics (MD) simulations

A 3D structural model of mutant FOXA1 (deletion of Phe254 and Glu255 henceforth referred to as FOXA1[F254E255]) complexed to DNA was generated using the model of the wild-type FOXA1[WT]–DNA complex that was published earlier (Arruabarrena-Aristorena et al, 2020). This complex was subject to atomistic molecular dynamics (MD) simulations using the same protocols that we had adopted to simulate the complex of FOXA1[WT] with DNA in the earlier study (Arruabarrena-Aristorena et al, 2020). The MD simulations were carried out with the pmemd.cuda module of the program Amber18 using the Amber ff14SB force field (Maier et al., 2015) for proteins and the amber force field FF99BSC0 (Pérez et al, 2007) for DNA. Three independent MD simulations (assigning different initial velocities) were carried out on the FOXA1[F254E255]–DNA complex for 100 ns each. To enhance the conformational sampling, the conformations of the FOXA1[WT]–DNA (taken from the previous study (Arruabarrena-Aristorena et al, 2020)) and FOXA1[F254E255]–DNA complexes at the end of the MD simulations were subjected to accelerated MD (aMD) (Pierce et al, 2012)

simulations as implemented in Amber18. aMD simulations were performed on both systems using the "dual-boost" version (Hamelberg et al, 2007). For the aMD simulations, the conventional MD simulations mentioned earlier were used to derive the required parameters (EthreshP, alphaP, EthreshD, alphaD). aMD simulations were carried out for 500 ns each. Simulation trajectories were visualized using VMD (Humphrey et al, 1996) and figures were generated using Pymol (De Lano, W., The PyMOL molecular graphics system. De Lano Scientific: San Carlos CA, USA, 2002).

## Statistical analysis

Data are represented as mean ± standard deviation (s.d.) of at least three independent biological replicates except when otherwise indicated. Differences were analyzed by Student's t test or one-way ANOVA with respectively Bonferroni's and Duncan's post-hoc corrections, using PRISM 6 (GraphPad V. 6.01). Differences in RNA and ChIP seq data were analyzed with Wald test followed by the Benjamini–Hochberg multiple test correction/default in DESeq2 or with the Mann–Whitney U Test as reported in figure legends. $P$-values < 0.05 were considered significant.

### Graphics
Figs. 1A, 4A, EV1A, EV2A, and synopsis were created with BioRender.com; Figs. 4 and EV4 were created with Inkscape.

## Data availability

RNA sequencing data have been deposited in the BioProject database under the accession number PRJNA1064118: https://dataview.ncbi.nlm.nih.gov/object/PRJNA1064118?reviewer=aa5gu5hcffhcp4ite50b2cdg3f. All other data supporting the findings of this study are available from the corresponding authors upon reasonable request.

The source data of this paper are collected in the following database record: biostudies:S-SCDT-10_1038-S44319-024-00335-y.

## Peer review information

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

## Acknowledgements

We thank current and former members of the Lunardi laboratory for experimental support and advice. We are grateful to all the staff at the CIBIO core facilities for technical assistance and support in data acquisition and analysis. Department CIBIO Core Facilities are supported by the European Regional Development Fund (ERDF) 2014–2020. This work was supported by The Giovanni Armenise-Harvard Foundation (Career Development Award), Italian Ministry of University and Research (PRIN 20174PLLYN), Associazione Italiana per la Ricerca sul Cancro (AIRC-IG 27893), Fondazione Trentina per la Ricerca sui Tumori (FTRT), and core funding from the Department CIBIO to AL; by the Associazione Italiana per la Ricerca sul Cancro (AIRC MFAG 2017-ID 20621) to AR; by Italian Association for Cancer Research (AIRC, MFAG-20344) and Worldwide Cancer Research (23-0321) to FC, by University of Trento (Starting Grants Young Researchers 2019) to AA, and by grants from the National Institutes of Health (R01 CA238005 and U01 CA261822) to MMS. Individual fellowships were awarded from the Fondazione Umberto Veronesi (FUV 2016) to AA, (FUV 2016-2017) to FCa, from the United States Department of Defence (W81XWH-18-1-0424) to FCa, from the University of Trento (Ph.D. fellowship) to DDF, VF, DD, and DB, and from the Pezcoller Foundation (Ph.D. fellowship) to EM. Finally, special thanks to Bruno Ravelli's friends.

## Author contributions

**Dario De Felice**: Conceptualization; Data curation; Formal analysis; Validation; Investigation; Visualization; Methodology; Writing—original draft; Project administration. **Alessandro Alaimo**: Data curation; Formal analysis; Investigation; Writing—original draft; Project administration; Writing—review and editing. **Davide Bressan**: Software; Formal analysis; Investigation; Visualization; Methodology; Writing—original draft; Writing—review and editing. **Sacha Genovesi**: Investigation; Project administration. **Elisa Marmocchi**: Investigation. **Nicole Annesi**: Investigation. **Giulia Beccaceci**: Investigation. **Davide Dalfovo**: Software; Investigation. **Federico Cutrupi**: Investigation. **Stefano Medaglia**: Investigation. **Veronica Foletto**: Investigation. **Marco Lorenzoni**: Investigation. **Francesco Gandolfi**: Investigation. **Srinivasaraghavan Kannan**: Software; Investigation; Visualization; Methodology. **Chandra S Verma**: Software; Investigation; Visualization; Methodology. **Alessandro Vasciaveo**: Software; Investigation. **Michael M Shen**: Resources; Funding acquisition; Writing—original draft. **Alessandro Romanel**: Resources; Software; Funding acquisition. **Fulvio Chiacchiera**: Conceptualization; Formal analysis; Funding acquisition; Visualization; Methodology; Writing—original draft. **Francesco Cambuli**: Conceptualization; Formal analysis; Supervision; Validation; Investigation; Methodology; Writing—original draft; Writing—review and editing. **Andrea Lunardi**: Conceptualization; Resources; Data curation; Formal analysis; Supervision; Funding acquisition; Validation; Visualization; Methodology; Writing—original draft; Project administration; Writing—review and editing.

Source data underlying figure panels in this paper may have individual authorship assigned. Where available, figure panel/source data authorship is listed in the following database record: biostudies:S-SCDT-10_1038-S44319-024-00335-y.

## Disclosure and competing interests statement

FCh is a consultant for Dompè Pharmaceuticals SPA (not related to this work). All other authors declare no competing interests.

# Expanded View Figures

**Figure EV1.   Induction of RA signaling-responsive genes and lumen formation by ATRA and DHT treatment in prostate organoids.**

(A) Schematic representation of the three main enzymatic steps of retinoid metabolism. (B) RNA-Seq analysis showing differentially expressed genes involved in the retinoid pathway upon single or combined administration of ATRA and DHT to mPrOs cultured in ENRA-- medium. Data are presented as mean value ± s.d. of $n = 3$ biological independent replicates. The indicated adjusted *p*-values were calculated with the Wald test followed by the Benjamini–Hochberg multiple test correction (default in DESeq2). (C) mPrOs (C57BL6/J-upper panel and CD1-lower panel) morphology after 6 days of administration of different concentration of ATRA. Scale bar, 1 mm. $N > 3$ independent biological replicates. (D) Phenotypic analysis of mPrOs cultured with 16 nM ATRA with or without DHT and Enzalutamide. Scale bar, 1 mm. $N = 3$ independent biological replicates.

▶

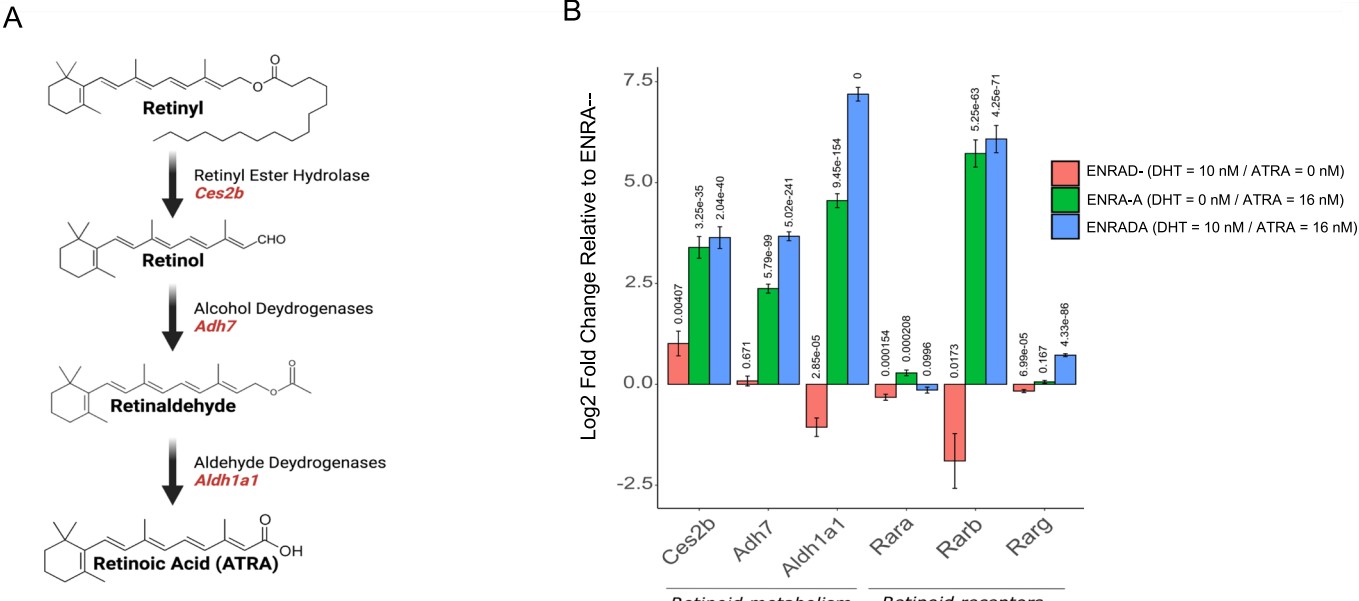

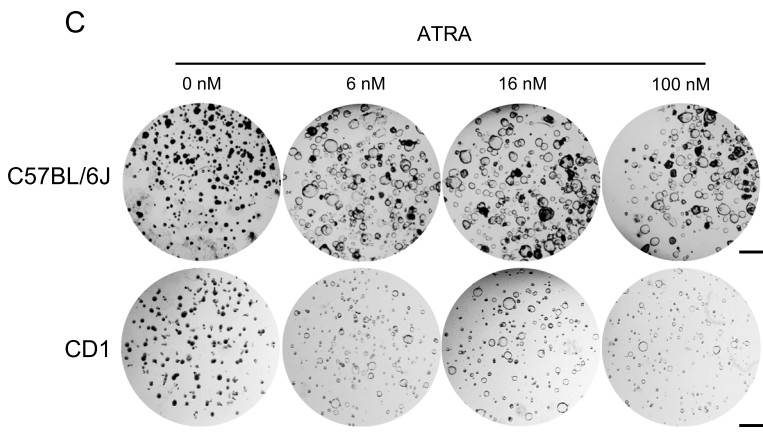

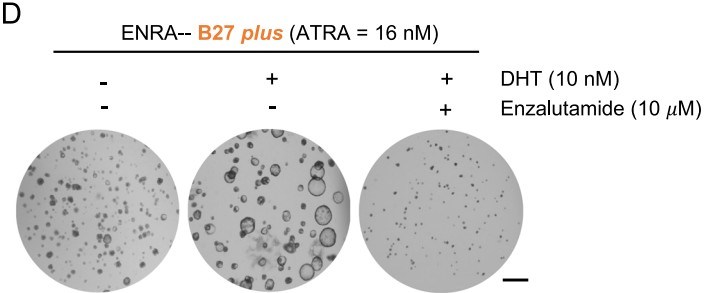

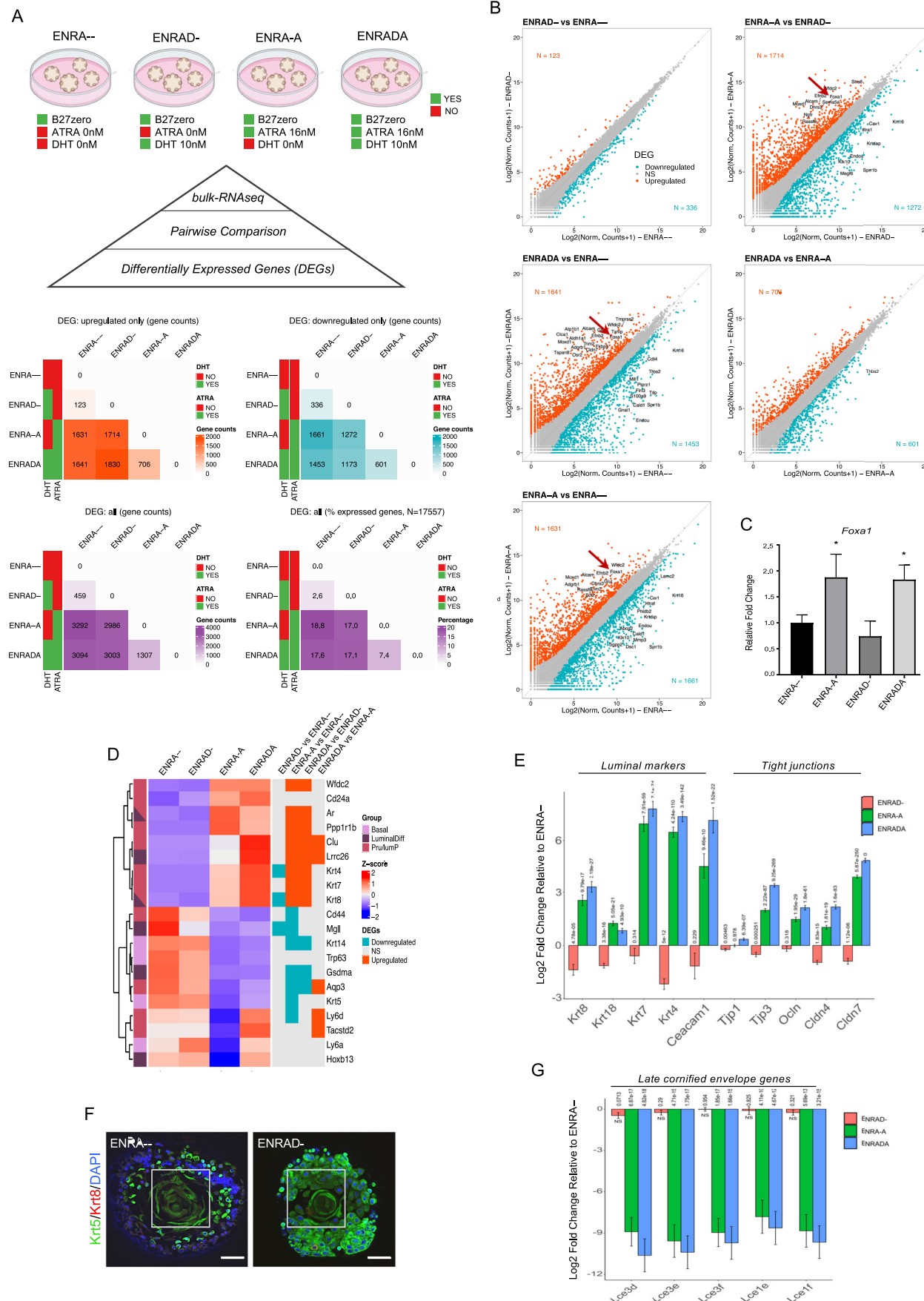

**Figure EV2. Transcriptional and phenotypic impact of ATRA treatment in prostate organoids.**

(A) Schematic representation of the cross-comparison study of RNA-seq analysis performed in this study (upper panels) and heatmaps displaying the number of DEGs (upregulated, downregulated, and total) for the 6 comparisons between the experimental conditions (ENRA--, ENRAD-, ENRA-A, ENRADA) (lower panels). The number indicated inside each cell of the matrix is a gene count except for the bottom right heatmap, which shows the percentage of DEGs over the total number of expressed genes ($N = 17,557$). Each cell number is the result of the differential expression analysis between the condition indicated in the row and the one in the column (e.g., 1631 is the number of upregulated genes in the comparison ENRA-A vs ENRA--). Upregulated (log2FC > 1) and downregulated (log2FC < −1) genes are the ones with an adj. *p*-value lower than 0.05 (Wald test followed by the Benjamini–Hochberg multiple test correction, default in DESeq2). (B) Scatter plots representing the changes in gene expressions in mPrOs grown for 6 days in indicated media. The number of significant up (log2FC > 1, orange) and down (log2FC < −1, light blue) regulated genes is indicated as N in the figure. Significance is assigned if the gene has an adj. *p*-value lower than 0.05 (Wald test followed by the Benjamini–Hochberg multiple test correction, default in DESeq2). Red arrow indicates Foxa1. (C) RT-qPCR analysis of Foxa1 gene expression in mPrOs kept for 5 day in ENRA-- and treated for 24 h with DHT, ATRA or the combination of both (Data are presented as mean value ± s.d. of $n = 3$ independent biological replicates, one-way ANOVA *$p = 0.022$). (D) Heatmap showing the expression of a selected panel of genes in mPrOs kept in the indicated culture conditions (mean of $n = 3$ biological independent replicates). Hierarchical clustering with average method has been applied on the heatmap rows. Genes are annotated as basal, luminal differentiated, and periurethral (PrU)/luminal progenitor (LumP) based on Crowley et al (2020) single-cell RNA sequencing analysis. Significant differentially expressed genes (DEGs) in the different comparisons are shown in red (upregulated) and turquoise (downregulated). Significance is assigned if the gene has an adj. *p*-value lower than 0.05 (Wald test followed by the Benjamini–Hochberg multiple test correction, default in DESeq2). (E) Differential expression of luminal marker and tight-junction genes in mPrOs grown under ENRADA, ENRA-A, ENRAD-, or ENRA-- culture conditions. Data are presented as mean value ± s.d. of $n = 3$ independent biological replicates. The indicated adjusted *p*-values were calculated with the Wald test followed by the Benjamini–Hochberg multiple test correction (default in DESeq2). (F) Immunofluorescence analysis of Krt5 and Krt8 in mPrOs cultured without DHT and ATRA (ENRA--) or with DHT only (ENRAD). The withe frame marks a peculiar cell morphology noticed only in the absence of ATRA. Scale bars, 100 μm. (G) RNA-seq bar plot representation of late cornified envelope genes (*LCE*) the expression of which is robustly repressed by RA signaling. Data are presented as mean value ± s.d. of $n = 3$ independent biological replicates. The indicated adjusted *p*-values were calculated with the Wald test followed by the Benjamini–Hochberg multiple test correction (default in DESeq2).

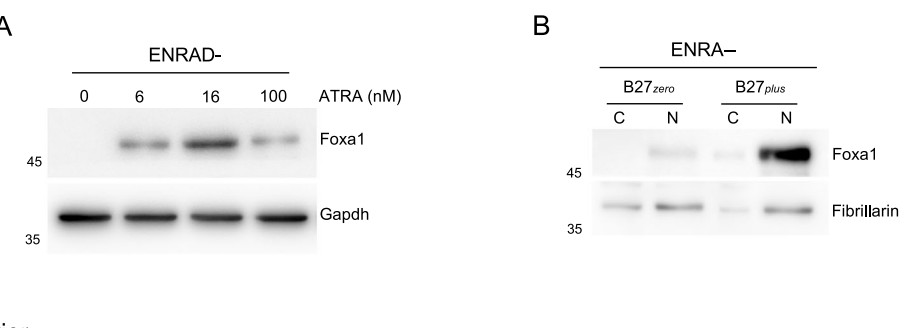

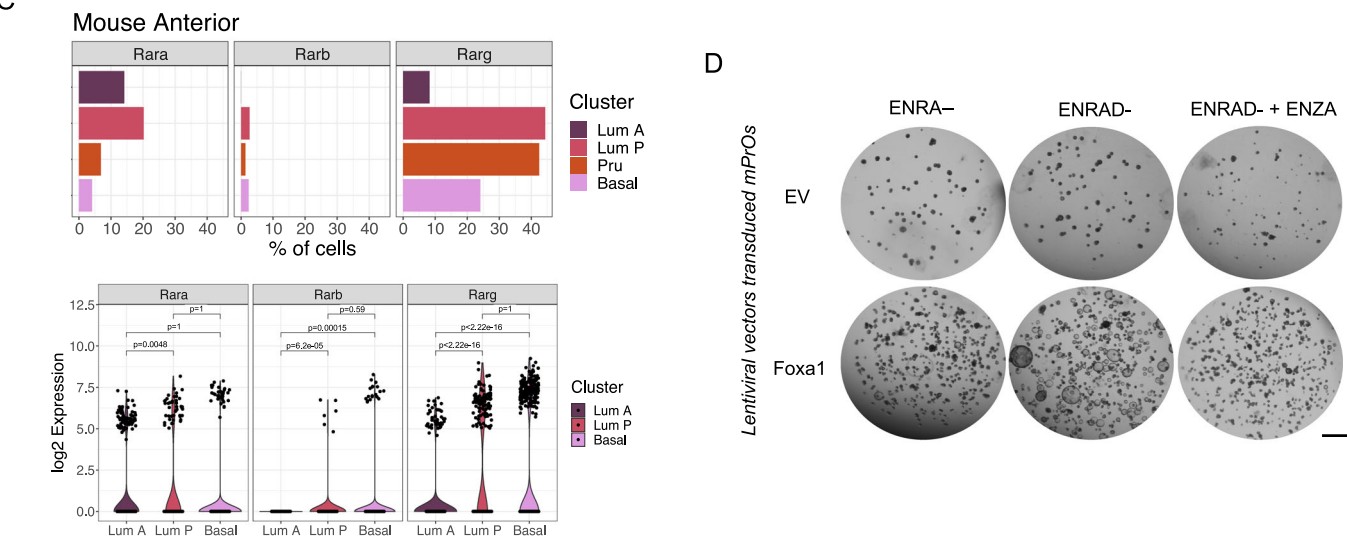

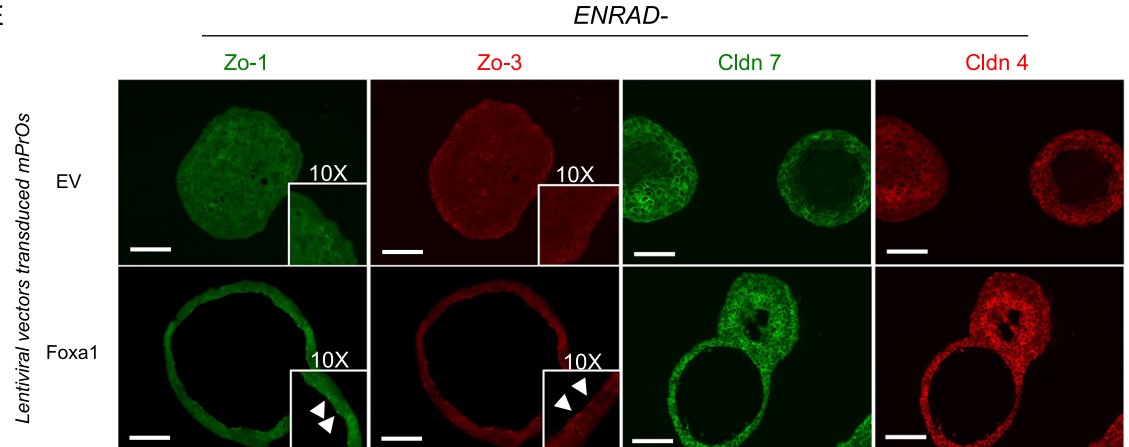

**Figure EV3. Molecular and phenotypic impact of modulating RA signaling, AR signaling, and Foxa1 expression in prostate organoids.**

(A) Western blot analysis of endogenous Foxa1 expression in mPrOs treated with different amounts of ATRA. Gapdh is used as loading control. $N = 2$ independent biological replicates. (B) Biochemical fractionation of cytosolic (C) and nuclear (N) compartments showing levels and localization of endogenous Foxa1 in the absence (B27$_{zero}$, ATRA 0 nM) or presence (B27$_{plus}$, ATRA 16 nM) of RA signaling. Fibrillarin is used as nuclear marker and loading control. (C) Percentage of cells (bar plots) and expression levels (violin plots) of, Rara Rarβ, and Rarγ genes in epithelial cell populations of mouse normal prostate (Data ref: Crowley et al, 2020; Appendix Fig. S2). The p-values indicated in the boxplots were calculated with the Mann–Whitney U Test. (D) Morphological analysis of transduced mPrOs ((empty vector (EV) and Foxa1)) cultured without ATRA and DHT (ENRA--), without ATRA with DHT (ENRAD-), and without ATRA with DHT plus Enzalutamide (ENZA 10 μM). Scale bar, 1 mm. (E) Immunofluorescence analysis of Zo-1 (Tjp1), Zo-3 (Tjp3), Cldn 4, and Cldn 7 expression and localization in transduced mPrOs (EV and Foxa1) cultured without ATRA but with DHT (ENRAD-). Magnification (10x) of Zo-1 and Zo-3 immunostaining are shown to pointing out protein localization. Nuclei are stained with DAPI. Scale bars, 100 μm. White arrowhead indicates Zo-1 and Zo-3 proteins localization. $N = 2$ independent biological replicates.

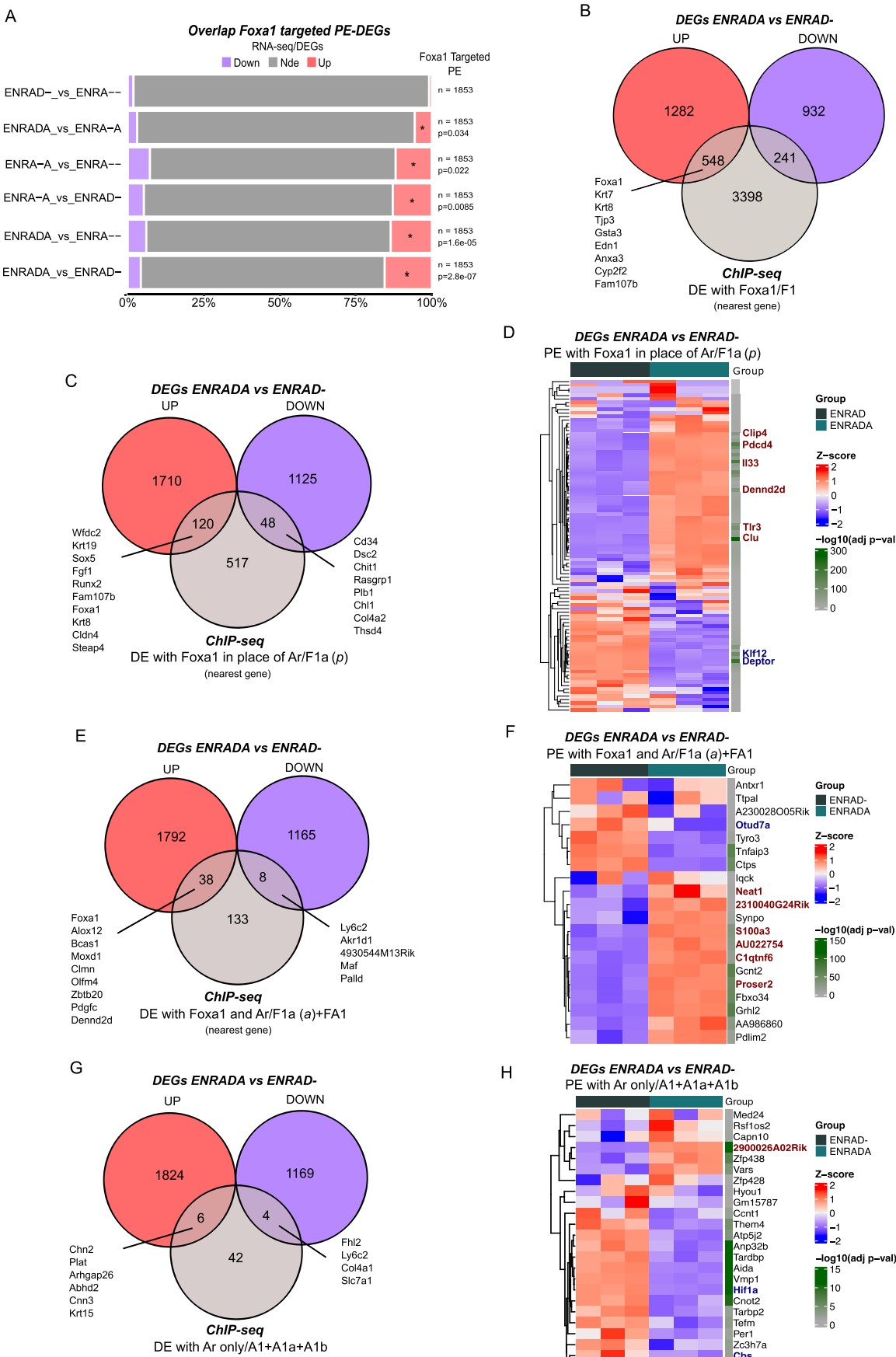

**Figure EV4.** **Transcriptional impact of retinoic acid and testosterone signaling in prostate organoids.**

(A) The barplot displays the overlap between differentially expressed genes (this work, comparison indicated on the left side) and all the Foxa1 PE-associated genes from ChIP-seq in the Foxa1 transgene expression condition (F1 + F1a + F1b + FA1). Genes are colored based on the RNA-seq status, i.e., upregulated (red), Downregulated (purple), or not differentially expressed (gray). The total number of FOXA1 PE-associated genes is 1853. The significance of the overlap between up- and down-regulated genes and the FOXA1 PE-associated genes has been determined by a hypergeometric test. The $p$-value is denoted by asterisks (***: 0–0.001, **: 0.001–0.01, *: 0.01–0.05, No symbol: 0.1–1.0). ChIP-seq data are from Adams et al, 2019. (B) Venn diagrams showing the overlap of differentially expressed genes in mPrOs cultured in ENRADA versus ENRAD, and exogenous Foxa1-bound distal elements (DE (F1)) in the genome. Relevant upregulated genes in the intersection are highlighted. (C) Venn diagram showing the overlap between DEGs in mPrOs cultured in ENRADA versus ENRAD and distal elements where exogenous Foxa1 replaces/displaces Ar. Relevant DEGs in the intersections are highlighted. (D) Heatmap showing DEGs in mPrOs cultured in ENRADA versus ENRAD on the promoter of which exogenous Foxa1 displaces/replaces Ar. The indicated adjusted $p$-values were calculated with the Wald test and then corrected with the Benjamini–Hochberg method (default method in DESeq2). Gene names highlighted in red indicate a significant upregulation (log2FC > 1, adj. $p$-value < 0.05), while gene names highlighted in blue indicate a significant downregulation (log2FC < −1, adj. $p$-value < 0.05). (E) Venn diagram showing the overlap between differentially expressed genes in mPrOs cultured in ENRADA versus ENRAD and distal elements concomitantly bound by both Ar and exogenous Foxa1. Relevant genes in the intersections are highlighted. (F) Heatmap showing DEGs in mPrOs cultured in ENRADA versus ENRAD whose promoter is concomitantly bound by both Ar and exogenous Foxa1. The indicated adjusted $p$-values were calculated with the Wald test and then corrected with the Benjamini–Hochberg method (default method in DESeq2). Gene names highlighted in red indicate a significant upregulation (log2FC > 1, adj. $p$-value < 0.05), while gene names highlighted in blue indicate a significant downregulation (log2FC < −1, adj. $p$-value < 0.05). (G) Venn diagram showing the overlap between DEGs in mPrOs cultured in ENRADA versus ENRAD and distal elements bound by Ar but not by exogenous Foxa1. Relevant genes in the intersections are highlighted. (H) Heatmap showing DEGs in mPrOs cultured in ENRADA versus ENRAD whose promoter is bound by Ar but not by exogenous Foxa1. The indicated adjusted $p$-values were calculated with the Wald test and then corrected with the Benjamini–Hochberg method (default method in DESeq2). Gene names highlighted in red indicate a significant upregulation (log2FC > 1, adj. $p$-value < 0.05), while gene names highlighted in blue indicate a significant downregulation (log2FC < −1, adj. $p$-value < 0.05).

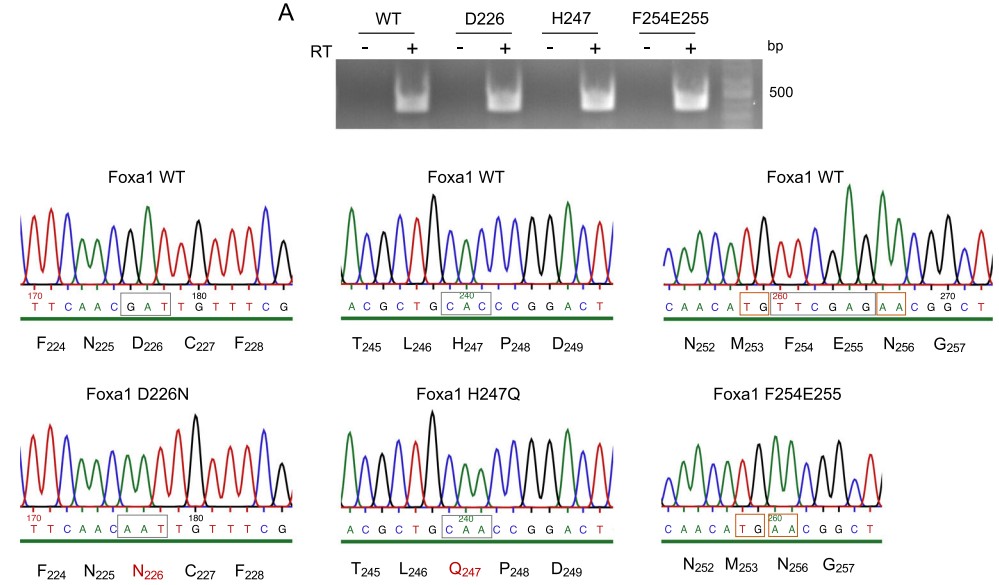

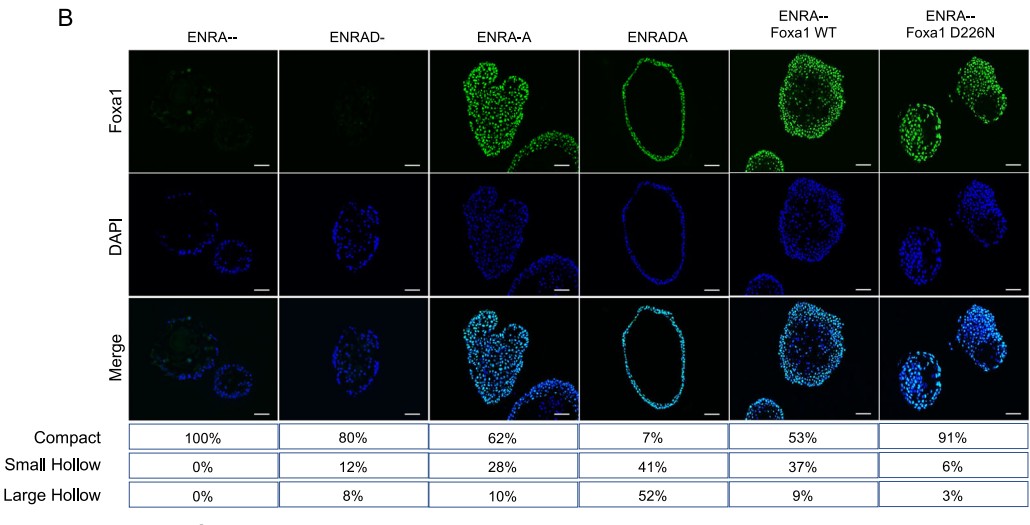

| | ENRA-- | ENRAD- | ENRA-A | ENRADA | ENRA--<br>Foxa1 WT | ENRA--<br>Foxa1 D226N |
|---|---|---|---|---|---|---|
| Compact | 100% | 80% | 62% | 7% | 53% | 91% |
| Small Hollow | 0% | 12% | 28% | 41% | 37% | 6% |
| Large Hollow | 0% | 8% | 10% | 52% | 9% | 3% |

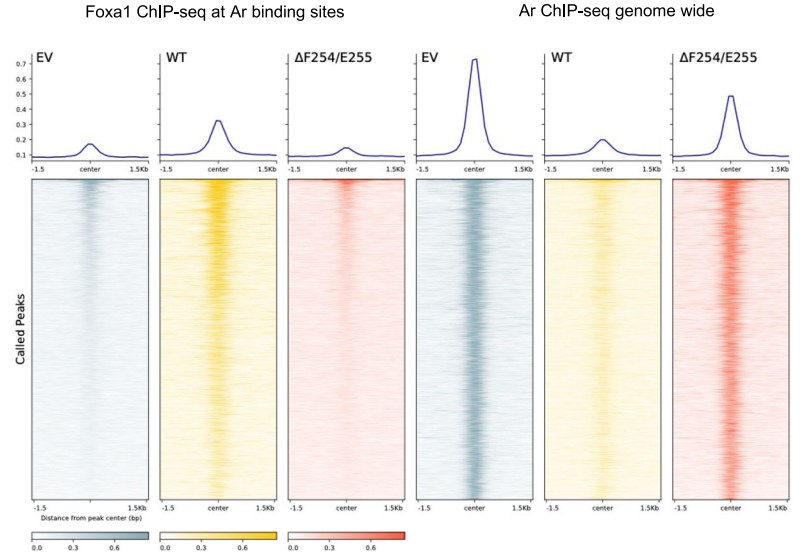

◄ **Figure EV5. Genetic engineering of prostate organoids with Foxa1 mutant isoforms.**

(A) RT-PCR and amplicons sequences of wild-type and mutant forms of Foxa1 stably expressed in mPrOs. Spectropherograms highlighting the mutated nucleotides in the different mPrOs lines. (B) Immunofluorescence analysis showing nuclear localization of endogenous and exogenous wild-type and mutant D226N Foxa1 in different growth culture conditions. Scale bar 50 μm. (C) Heatmap showing the signal intensity of Foxa1 and Ar binding over AR genome-wide binding sites (ChIP-seq from Adams et al, 2019). ChIP-seq was performed on mPrOs stably transduced with wild-type Foxa1, Foxa1$^{F254E255}$, or the empty vector (EV) and cultured with DHT but not ATRA.

