## [Peer Review File · EMBO Reports]

Rarg-Foxa1 signaling promotes luminal identity in prostate progenitors and is disrupted in prostate cancer

Dario De Felice, Alessandro Alaimo, Davide Bressan, Sacha Genovesi, Elisa Marmocchi, Nicole Annesi, Giulia Beccaceci, Davide Dalfovo, Federico Cutrupi, Stefano Medaglia, Veronica Foletto, Marco Lorenzoni, Francesco Gandolfi, Srinivasaraghavan Kannan, Chandra Verma, Alessandro Vasciaveo, Michael Shen, Alessandro Romanel, Fulvio Chiacchiera, Francesco Cambuli, and Andrea Lunardi

Corresponding author(s): Andrea Lunardi (andrea.lunardi@unitn.it), Francesco Cambuli (francesco.cambuli@fht.org)

Review Timeline:

Submission Date:	16th Apr 24
Editorial Decision:	12th Jun 24
Revision Received:	9th Sep 24
Editorial Decision:	16th Oct 24
Revision Received:	6th Nov 24
Accepted:	14th Nov 24

Editor: Esther Schnapp

Transaction Report:

Dear Andrea,

Thank you for the submission of your manuscript to EMBO reports. We have now received the full set of referee reports that is pasted below.

As you will see, the referees acknowledge that the findings are potentially interesting. Together, they only have some suggestions for how the manuscript could be improved. I think all of them should be addressed, except point 3 by referee 3 does not have to be addressed experimentally (identifying additional RA targets in the pathway). It would be good if the link between the FOXA1 prostate cancer mutant, lack of luminal lineage commitment and cancer formation could be made clearer. If these links are unknown, please do discuss this point.

Please let me know if you have any comments or questions regarding the revisions, and we can discuss this further, also in a video chat, if you like.

I would thus like to invite you to revise your manuscript with the understanding that the referee concerns must be fully addressed and their suggestions taken on board. Please address all referee concerns in a complete point-by-point response. Acceptance of the manuscript will depend on a positive outcome of a second round of review. It is EMBO reports policy to allow a single round of major revision only and acceptance or rejection of the manuscript will therefore depend on the completeness of your responses included in the next, final version of the manuscript.

We realize that it is difficult to revise to a specific deadline. In the interest of protecting the conceptual advance provided by the work, we recommend a revision within 3 months (12th Sep 2024). Please discuss the revision progress ahead of this time with the editor if you require more time to complete the revisions.

- 1) A data availability section providing access to data deposited in public databases is missing. If you have not deposited any data, please add a sentence to the data availability section that explains that.
- 2) Your manuscript contains statistics and error bars based on $n=2$. Please use scatter blots in these cases. No statistics should be calculated if $n=2$.

3) We replaced Supplementary Information with Expanded View (EV) Figures and Tables that are collapsible/expandable online. A maximum of 5 EV Figures can be typeset. EV Figures should be cited as 'Figure EV1, Figure EV2' etc... in the text and their respective legends should be included in the main text after the legends of regular figures.

4) a .docx formatted letter INCLUDING the reviewers' reports and your detailed point-by-point responses to their comments. As part of the EMBO Press transparent editorial process, the point-by-point response is part of the Review Process File (RPF),

which will be published alongside your paper.

5) a complete author checklist, which you can download from our author guidelines

<<https://www.embopress.org/page/journal/14693178/authorguide>>. Please insert information in the checklist that is also reflected in the manuscript. The completed author checklist will also be part of the RPF.

6) Please note that all corresponding authors are required to supply an ORCID ID for their name upon submission of a revised manuscript (<<https://orcid.org/>>). Please find instructions on how to link your ORCID ID to your account in our manuscript tracking system in our Author guidelines

<<https://www.embopress.org/page/journal/14693178/authorguide#authorshipguidelines>>

10) Regarding data quantification (see Figure Legends:

<https://www.embopress.org/page/journal/14693178/authorguide#figureformat>)

- the name of the statistical test used to generate error bars and P values,
- the number (n) of independent experiments (please specify technical or biological replicates) underlying each data point,
- the nature of the bars and error bars (s.d., s.e.m.),
- If the data are obtained from $n < 3$, use scatter blots showing the individual data points.

12) All Materials and Methods need to be described in the main text. We would encourage you to use 'Structured Methods', our new Methods format. According to this format, the Methods section should include a Reagents and Tools Table (listing key reagents, experimental models, software and relevant equipment and including their sources and relevant identifiers) followed by a Methods and Protocols section in which we encourage the authors to describe their methods using a step-by-step protocol format with bullet points, to facilitate the adoption of the methodologies across labs. More information on how to adhere to this format as well as downloadable templates (.docx) for the Reagents and Tools Table can be found in our author guidelines: < <https://www.embopress.org/page/journal/14693178/authorguide#manuscriptpreparation>>.

An example of a Method paper with Structured Methods can be found here: <https://www.embopress.org/doi/full/10.1038/s44320-024-00037-6#sec-4>

I look forward to seeing a revised form of your manuscript when it is ready.

Referee #1:

EMBOR-2024-59431V1

Rarg -Foxa1 signaling promotes luminal identity in prostate progenitors and is disrupted in prostate cancer

The study of Felice et al., focused on the role of retinoid signaling on prostate luminal epithelium specification, and showed its interaction with of the most important transcription factors in prostate tissue regulation and carcinogenesis, FOXA1. The authors conducted thorough examination of cell culture conditions and their impact on cell signaling, which is a great example of how specific components and their dosage used should be reported in studies, given that they can greatly alter cell phenotype. They uncovered that retinoic acid nuclear receptor RAR γ alters the FOXA1 and subsequently AR occupation of regulatory binding sites, thus affecting gene expression of luminal lineage specification genes. The key finding of FOXA1 governing luminal specification via direct binding to regulatory elements, completely independently of AR the main inducer of luminal fate is of particular interest to the field, for molecular mechanism elucidation in prostate cancer.

1. Related to novelty, the axis RA - Foxa1 has been previously identified in developmental biology, such as in embryonic stem cell systems. Please address this in your discussion and specify the new findings from your study on prostate luminal identity regulation.
2. The modulation of AR binding sites in the FOXA1 F254 mutant is quite interesting, however FOXA1 indels in the C terminal area seems to be impacting only a small number of patients (Fig. 4A-B), thus questioning its clinical relevance. Please comment and also provide additional information on the stage of PCa and the prevalence of these mutations in the mentioned figure panel.
3. Can the luminal specification loss, observed by the F254 mutant, be rescued via activation of RAR γ ? I would recommend using the established in vitro modelling of specific mutants in organoids to answering whether stimulation with exogenous RA would (partially) revert the occupation of binding sites, which would complement the analysis shown in Fig4. G-K and S5.
4. The quantification of compact/ hollow and luminal organoid structures should be done in S5B.

Referee #2:

1. In the introduction it's a little unclear why focus on RAR of all three RARs, and potentially, given the existing literature, the authors could refine their question further to justify the focus and the interactions with FOXA1. (PMID:36768694; PMID: 32881426; PMID: 30120411; PMID: 22362749; PMID:38428412; PMID: 36052494; PMID: 35802768; PMID: 24492483; PMID: 19623543; PMID: 15651062; PMID: 15217932; PMID: 10872810; PMID: 10459851; PMID: 9528984; PMID: 9075707; PMID: 38168185)
2. In intro (and ms) use correct case for murine and human gene studies, e.g. lineage tracing experiments are in animals? And case should be corrected

3. P.5/6 culture optimization, whilst interesting, could be considerably condensed (or placed in supplemental information)
4. By contrast, the introduction of the RNA-Seq experiment is brief, and actually is quite complex. If this reviewer understands correctly there are four conditions and it's not always clear what comparisons are being undertaken. Indeed the figure 2A seems an unusual way to illustrate some of these comparisons. For example, in this figure it's not clear if these are differentially expressed genes (it seems not?) so there are the same genes listed on both sides of the correlation plot and the reader is aiming to compare them? By contrast it would be more helpful to have a multipaneled volcano plot (4x4?) that shows the matrix of comparisons of differentially expressed genes, with the top n genes labelled, and then the differential expression of Foxa1 could be more precisely gauged; currently, F2A doesn't really justify investigating Foxa1.
5. In general, the ENRADA etc nomenclature is a little confusing, and would be better to be spelt out.
6. How many, and which, Fox family members were regulated in the RNA-Seq?
7. Supp 2 - unclear why select these genes, and also, in each comparison how did these genes rank? It's also a little unusual to visualize these as normalized counts instead of Z scores, and also means the data are less easy to interpret
8. GSEA or other network analyses should be undertaken. It's perhaps a little misleading to select genes to justify phenotype responses without a more agnostic layer of analyses to put these findings in context.
9. The BioProject # PRJNA1064118 doesn't return a record
10. Figure 2A-D - unclear what time point of exposure is used, and should be stated, as this will be indicative of whether this is a direct or indirect event; for example, Foxa1 kd impeding RA functions is consistent with both direct and indirect effects
11. ChIP-Seq and RNA-Seq integration. Unclear what distance was used to annotate to genes and should be indicated (e.g. Supp F4 - the overlap will depend on the distance) Also, the method seems to suggest only up-regulated genes were used, but in the RNA-Seq data there are significant up- and down-regulated genes.
12. It's indicated on p.8 that the FoxA1 ChIP-Seq was reanalyzed. Why? Did the reanalysis reflect what was published in the Adams 2019 study?
13. Intersection of RNA-Seq and ChIP-Seq - given that there is potentially a grid of multiple comparisons (see #4), where is the Foxa1 signal most enriched?
14. Supp Tabel S2 - apologies, I don't see the supp tables, where do the luminal etc markers rank in the genes (for example by considering strength of Foxa1 enrichment and magnitude of regulation?)
15. The text on p.8 and p.9 is quite dense, and there's no significance associated with any of the statements (although the methods suggest a hypergeometric test was done). For example, it's unclear how statements about replaced and joined are made as there's only one time point and there's no kinetic data
16. The introduction to FOXA1 mutations on p.10 would be better in the introduction
17. S4 - the nomenclature on the Venn diagrams is unclear to this reviewer
18. Unclear rationale for the study of the Fox mutations, as it doesn't obviously relate the RARg or ATRA signaling as developed in the previous experiments, and somewhat overlaps with the Adams et al paper. Apologies if this is misunderstood

Referee #3:

In this manuscript, Dario De Felice and collaborators shed light on Retinoic acid (RA) as a key signal regulator for glandular identity in adult prostate progenitors and suggest that disruptions in vitamin A metabolism may be a risk factor for prostate diseases. The scientific article focuses on the critical roles of Vitamin A (retinol) in embryonic development, organogenesis, and tissue differentiation. It explores the mechanisms of action of Vitamin A, specifically through its active derivative, retinoic acid (RA), and its receptors (RARa, RARb, and RARg). The study highlights the significance of RA-RARg signaling in prostate development, particularly through the target gene Foxa1, which is essential for luminal lineage differentiation and proper prostate function.

Key findings include i) Foxa1's role in shaping androgen signaling and promoting the expression of genes critical for luminal progenitor cells; ii) Challenges in modeling the luminal compartment in vitro, with current organoid systems lacking robust secretion of kallikrein-rich fluids; iii) Identification of FOXA1 as a major target of RA signaling, with implications for prostate cancer, where FOXA1 mutations may enhance transcriptional activity and contribute to tumorigenesis.

The study emphasizes the need for further research on the molecular mechanisms regulated by RA and FOXA1, particularly in the context of cancer treatment, where retinoid derivatives may offer therapeutic potential for tumors with dysfunctional RA signaling and specific FOXA1 mutations.

This study offers new insights into the signaling pathways and molecular circuits that regulate prostate progenitor commitment and adult tissue homeostasis. The article provides an extensive overview of Vitamin A's role in embryonic development, organogenesis, and tissue differentiation, highlighting the nutrient's critical importance in these processes. The identification and characterization of retinoic acid (RA) and its receptors (RARa, RARb, RARg) are thoroughly detailed, providing significant insights into their distinct expression patterns and functions. The use of mouse prostate organoids to study RA signaling and its impact on Foxa1 offers a novel and practical approach to understanding the molecular mechanisms underlying prostate development.

Similarly, it paves the way for more precise assessments of retinoid derivatives in treating prostate tumors. Exploring FOXA1 mutations in prostate and breast cancers and their impact on epithelial transformation and tumorigenesis provides valuable implications for cancer research and potential therapeutic strategies. Hence, this work will undoubtedly be of general interest to

the readers of EMBO Reports.

However, there are some minor weaknesses to consider that should be addressed before the publication of this manuscript:

1- While the use of organoid systems is innovative, the study acknowledges that these systems currently lack robust and reproducible secretion of kallikrein-rich fluids, limiting the model's physiological relevance. This issue should be more clearly stated in the discussion.

2- Also, the article points out that the molecular circuits underlying the processes influenced by RA signaling and Foxa1 remain understudied, indicating a gap in the comprehensive understanding of these pathways.

3- Although the study provides significant insights, the exact molecular mechanisms by which Foxa1 and RA signaling regulate luminal progenitor fate and prostate function are not fully elucidated. While Foxa1 is identified as a major target, the author's study may explore additional targets and pathways involved in RA signaling to provide a more holistic view of its role in prostate development and cancer.

4- Merge in Fig. 5(F) is advised.

5- The article could strengthen its impact by outlining more specific future research directions or experimental approaches to address the identified gaps and weaknesses.

6- A general review of abbreviations is advised (see pg.3 AR).

EMBOR-2024-59431V1

We thank our Reviewers for the insightful comments and suggestions that helped strengthen the robustness of our results and improve the soundness of our conclusions.

Below is the point-by-point rebuttal to the Referees' comments:

Referee #1:

RAR γ -Foxa1 signaling promotes luminal identity in prostate progenitors and is disrupted in prostate cancer

The study of De Felice et al., focused on the role of retinoid signaling on prostate luminal epithelium specification, and showed its interaction with of the most important transcription factors in prostate tissue regulation and carcinogenesis, FOXA1. The authors conducted thorough examination of cell culture conditions and their impact on cell signaling, which is a great example of how specific components and their dosage used should be reported in studies, given that they can greatly alter cell phenotype. They uncovered that retinoic acid nuclear receptor RAR γ alters the FOXA1 and subsequently AR occupation of regulatory binding sites, thus affecting gene expression of luminal lineage specification genes. The key finding of FOXA1 governing luminal specification via direct binding to regulatory elements, completely independently of AR the main inducer of luminal fate is of particular interest to the field, for molecular mechanism elucidation in prostate cancer.

We thank the Reviewer for recognizing the relevance of our work.

1. Related to novelty, the axis RA - Foxa1 has been previously identified in developmental biology, such as in embryonic stem cell systems. Please address this in your discussion and specify the new findings from your study on prostate luminal identity regulation.

R. We thank the Reviewer for her/his suggestion. The following paragraph has been integrated in the Discussion.

"The ability of RA signaling to promote cell lineage commitment via FOXA1 (alias HNF-3 α) was previously reported in the context of embryonal development (e.g., neuronal tissue, endoderm) (Jacob et al, 1994, 1999; Tan et al, 2010; Taube et al, 2010). Recently, the vitamin A metabolite Retinoic Acid (RA) has been shown to restrict the lineage plasticity of adult stem cells of the skin (Tierney et al, 2024), shedding light on a pivotal role of RA signaling in adult progenitor lineage commitment, and its deregulation during tumorigenesis."

2. The modulation of AR binding sites in the FOXA1 F254 mutant is quite interesting, however FOXA1 indels in the C terminal area seems to be impacting only a small number of patients (Fig. 4A-B), thus questioning its clinical relevance. Please comment and also provide additional information on the stage of PCa and the prevalence of these mutations in the mentioned figure panel.

R. We thank the Reviewer to rise this point. According to cBioPortal, the incidence of FOXA1 mutation in PCa is about 11%, one quarter of which is in the c-terminal part of the Forkhead domain (Figure R1). Of the latter, 16% involve the phenylalanine residue at position 254 (F254) and are almost invariably indels. Because the analysis in cBioPortal has been done by pooling 28 different datasets, information about the stage of the tumors carrying the F254 mutation is very difficult to retrieve. According to GloboCan, 1.467.854 new cases of PCa were diagnosed worldwide in the 2022, of which 6.458 were presumably characterized by the mutation of FOXA1 residue F254.

3. Can the luminal specification loss, observed by the F254 mutant, be rescued via activation of RARg?

I would recommend using the established in vitro modelling of specific mutants in organoids to answering whether stimulation with exogenous RA would (partially) revert the occupation of binding sites, which would complement the analysis shown in Fig 4. G-K and S5.

R. We thank our Reviewer for her/his suggestion. As shown in Figure R2, in presence of DHT, ATRA induces the formation of organoids in the F254E255 Foxa1 mutant line. The rescue is very likely dependent by the atRA-dependent induction of endogenous wild type Foxa1 expression in mPrOs (new Figure EV2C, 16 nM ATRA, 24h), which suggests the lack of a dominant negative effect of F254E255 mutant Foxa1 on the endogenous wild-type counterpart. Unfortunately, to avoid possible interference with the transcriptional activity of Foxa1, exogenous Foxa1 proteins do not have tags in their C- or N-terminal regions, which makes it impossible discriminate exogenous Foxa1 protein from endogenous ones by western blot as well as DNA binding of the endogenous Foxa1 protein from that of the exogenous Foxa1 protein by ChIP experiments. The generation of more appropriate models and the development of dedicated molecular analyses are therefore needed to adequately address this interesting issue, which therefore cannot be addressed in the timeframe given for the revision.

4. The quantification of compact/ hollow and luminal organoid structures should be done in S5B.

R. We thank the Reviewer for her/his suggestion. Percentage of compact, small hollow and large hollow organoids is now included in Figure EV5B.

R. We thank the Reviewer for her/his suggestion. Percentage of compact, small hollow and large hollow organoids is now included in Figure EV5B.

Referee #2:

1. In the introduction it's a little unclear why focus on RAR γ of all three RARs, and potentially, given the existing literature, the authors could refine their question further to justify the focus and the interactions with FOXA1. (PMID:36768694; PMID: 32881426; PMID: 30120411; PMID: 22362749; PMID:38428412; PMID: 36052494; PMID: 35802768; PMID: 24492483; PMID: 19623543; PMID: 15651062; PMID: 15217932; PMID: 10872810; PMID: 10459851; PMID: 9528984; PMID: 9075707; PMID: 38168185)

R. We thank our Reviewer for her/his suggestion. We emphasized Rar γ in the Introduction mainly for its potential role in mouse prostate progenitors and their commitment towards a secretory epithelium. Since we have always considered wild type mouse prostate organoids as a beautiful model of healthy adult prostate progenitors, our initial hypothesis was to investigate the role of RA signaling in adult prostate progenitors by exploiting the mPrOs technology without bias for any of the Rars or Foxa1. The results of different analyses focused our attention first on Foxa1 and then on Rar γ .

However, we agree with our Reviewer that it is important to emphasize in the introduction the special relevance of RAR γ over other RARs for prostate pathophysiology, not only for its role in the establishment and maintenance of the luminal compartment of the gland, but also considering its functional interaction with such a crucial pathway as androgen signaling in prostate cancer.

The following paragraph has been included in the Introduction of the revised manuscript.

“Retinoids are vitamers of Vitamin A involved in many biochemical processes, including cell differentiation and embryonic development (Petkovich & Chambon, 2022). The pleiotropic actions of retinoids are mediated by two families of nuclear receptors: retinoic acid receptors (RARs) α , β , and γ and retinoid X receptors (RXRs) (Heyman et al, 1992; Levin et al, 1992; Allenby et al, 1993). Upon RA binding, these receptors recognize complementary DNA elements in the regulatory regions of selected genes and modulate their expression (Evans, 1988; Beato, 1989). Specifically, RAR γ has been shown to be deregulated in prostate cancer and to influence the androgen receptor cistrome in malignant cell lines (Long et al, 2019; Petrie et al, 2020; Bhowmick & Bhowmick, 2022; Yu et al, 2022; Wani et al, 2023). Yet, understanding the impact of specific signaling events on prostate self-renewal and differentiation requires control over microenvironmental conditions and genetic perturbations, which is more difficult to achieve in conventional cancer cell lines adapted to grow in serum-rich media.”

2. In intro (and ms) use correct case for murine and human gene studies, e.g. lineage tracing experiments are in animals? And case should be corrected

R. We thank the Reviewer for this suggestion. All gene are reported according to the current nomenclature (e.g. *Foxa1* mouse gene, *Foxa1* mouse RNA and protein; *FOXA1* human gene, *FOXA1* human RNA and protein).

3. P.5/6 culture optimization, whilst interesting, could be considerably condensed (or placed in supplemental information)

R. We agree with our Reviewer that the initial section of the Results is quite detailed, but the field of prostate organoids is often characterized by summarily described experiments that make difficult the interpretation of the results. For this reason, we consider details very relevant for this work.

4. By contrast, the introduction of the RNA-Seq experiment is brief, and actually is quite complex. If this reviewer understands correctly there are four conditions and it's not always clear what comparisons are being undertaken.

Indeed, the figure 2A seems an unusual way to illustrate some of these comparisons.

For example, in this figure it's not clear if these are differentially expressed genes (it seems not?) so there are the same genes listed on both sides of the correlation plot and the reader is aiming to compare them?

R. We apologize with our Reviewer if the analysis in Figure 2A was not sufficiently clear. Fig 2A shows the differentially expressed genes as colored (up-regulated in orange; down-regulated in light blue). These genes have a $\log_2FC > 1$ (up) or < -1 (down) and $\text{adj. } P\text{-value} < 0.05$. We identified *Foxa1* among the top up-regulated hits in the comparisons ENRA+ATRA vs ENRA, ENRA+ATRA vs ENRA+DHT, and ENRA+DHT+ATRA vs ENRA+DHT. These comparisons point out the dependence of *Foxa1* expression on RA signaling. For clarity, all the comparisons are now shown as Figure EV2B in the revised manuscript. However, to better clarify the analysis we included the new panel A in Figure EV2 describing the experimental setting and results.

By contrast it would be more helpful to have a multipaneled volcano plot (4x4?) that shows the matrix of comparisons of differentially expressed genes, with the top n genes labelled, and then the differential expression of Foxa1 could be more precisely gauged; currently, F2A doesn't really justify investigating Foxa1.

R. We thank the Reviewer for her/his suggestion. Volcano plots are now showed in the new Appendix Figure S2 of the revised manuscript.

5. In general, the ENRADA etc nomenclature is a little confusing, and would be better to be spelt out.

R. We apologize with our Reviewer if the nomenclature appear a bit confusing. The nomenclature of mouse prostate organoid growth media is aligned with that defined by the groundbreaking works published in 2014 by the groups of Hans Clevers and Charles Sawyers (Karthaus et al., 2014 *Cell*; Drost et al., 2016 *Nat Prot*). Over the past decade, these papers have been a benchmark for the field, therefore we feel it is important to maintain the very same nomenclature (ENRA=EGF+Noggin+R-Spondin+A-8301; ENRAD=ENRA+DHT; ENRADA=ENRA+DHT+atRA; ENRA-A=ENRA+atRA). Of note, DHT and ATRA are the only two factors that change in the different growth conditions while EGF, Noggin, R-Spondin, and A-8301 (ENRA) are always present in the media.

6. How many, and which, Fox family members were regulated in the RNA-Seq?

R. We agree with our Reviewer that an unbiased analysis of the members of the Fox family of transcription factors can be very interesting in our experimental setting. Volcano plots showed in the new Appendix Figure S2 of the revised manuscript address this point and highlight *Foxa1* gene as the most responsive member of the family to RA signaling.

7. Supp 2 - unclear why select these genes, and also, in each comparison how did these genes rank? It's also a little unusual to visualize these as normalized counts instead of Z scores, and also means the data are less easy to interpret.

R. We thank the reviewer for her/his suggestion and apologize if it was not sufficiently clear why we analyzed specific genes. The genes were selected from a single-cell atlas of the mouse prostate (Crowley et al. 2020) as markers of specific cell types, progenitors or differentiated prostate epithelial cells. We modified the heatmap to show z-scores and included an annotation to indicate whether the gene was up or down-regulated in the different conditions of our experiment. The new panel is now part of the revised version of the manuscript as Figure EV2D.

8. GSEA or other network analyses should be undertaken. It's perhaps a little misleading to select genes to justify phenotype responses without a more agnostic layer of analyses to put these findings in context.

R. We agree with the Reviewer that a network analysis may help to better define the phenotypes associated with the presence of androgen and retinoic signals. GSEA results have been anticipated in the Results section of the revised manuscript and presented as Appendix Figure S1 and Table EV1.

9. The BioProject # PRJNA1064118 doesn't return a record

R. We apologize with our Reviewer for the inconvenience. Here is the link that allow Reviewer to access BioProject and associated SRA metadata

<https://dataview.ncbi.nlm.nih.gov/object/PRJNA1064118?reviewer=aa5gu5hcffhcp4ite50b2cdg3f>

10. Figure 2A-D - unclear what time point of exposure is used, and should be stated, as this will be indicative of whether this is a direct or indirect event; for example, *Foxa1* kd impeding RA functions is consistent with both direct and indirect effects.

R. We agree with the Reviewer that experimental time points can help better decipher the molecular mechanism linking RA-RAR γ signaling to *Foxa1* induction. To obtain well-shaped organoids inside the Matrigel domes are generally required five/six days of culture this is the reason why we have almost invariably used this time point in our experiments. The time point is now included in the legends of Figure 2 A-D. To support the thesis of a direct role of RA-RAR γ signaling on the induction of *Foxa1* gene expression we have included the panel in Figure EV2C showing *Foxa1* expression upon 24 hours of ATRA or DHT treatment.

11. ChIP-Seq and RNA-Seq integration. Unclear what distance was used to annotate to genes and should be indicated (e.g. Supp F4 - the overlap will depend on the distance). Also, the method seems to suggest only up-regulated genes were used, but in the RNA-Seq data there are significant up- and down-regulated genes.

R. Apologies, we included the distance used in the Methods but not in the figure legend. For each peak (enhancer or promoter), we selected the nearest TSS as its target gene. Moreover, a peak was annotated as a promoter if it was in a window of ± 2.5 kb from a TSS. Thank you for pointing this out. We fixed the method section to specify that we used up and down-regulated genes.

12. It's indicated on p.8 that the *FoxA1* ChIP-Seq was reanalyzed. Why? Did the reanalysis reflect what was published in the Adams 2019 study?

R. We decided to reanalyze the data published in the 2019 Adams study because it is our opinion that the ChIP seq analysis shown in Figure 3A, 4C and luciferase experiments of 4E shown in Adams et al., 2019 do not support the gain-of-function phenotype attributed to *Foxa1* F254E255 mutant. Our genome-wide analyses show a substantial reduction in DNA binding of *Foxa1* mutant F254E255 compared to wild type *Foxa1*. A similar reduction in binding of *Foxa1* mutant F254E255 compared with *Foxa1* wild type is also evident on *Ar* binding sites. To reinforce these data, the occupancy of *Ar* binding sites by *Ar* in the presence of *Foxa1* mutant F254E255 is substantially higher than that in the presence of *Foxa1* wild type and comparable to the empty vector condition.

13. Intersection of RNA-Seq and ChIP-Seq - given that there is potentially a grid of multiple comparisons (see #4), where is the *Foxa1* signal most enriched?

R. We agree with our Reviewer that showing *Foxa1* signal across the different comparison can strength our conclusions. Below you can see the new graph, now included in the revised version of the manuscript as Figure EV4A, showing the overlap between *Foxa1* targeted PE and DEGs across the different comparisons.

14. Table EV2 - apologies, I don't see the supp tables, where do the luminal etc markers rank in the genes (for example by considering strength of *Foxa1* enrichment and magnitude of

regulation?).

R. Based on the new Figure EV4, we analyzed the magnitude of regulation of different luminal markers in the comparison ENRADA vs ENRAD. Genes are ranked according to decrescent pValue as per the new Table EV3:

#1. Tmprss2 = Upregulated, log2FC = 3.6, pValue = 0
#7. Wfdc2 = Upregulated, log2FC = 4.7, pValue = 0
#10. Cldn7 = Upregulated, log2FC = 5,7, pValue = 0
#44. Foxa1 = Upregulated, log2FC = 4,2, pValue = 0
#72. Tjp3 = Upregulated, log2FC = 3.9, pValue = 1.2E-299
#95. Clu = Upregulated, log2FC = 3,2, pValue = 6.9E-271
#129 Krt4 = Upregulated, log2FC = 9,5, pValue = 4,2E-228
#213 Cldn4 = Upregulated, log2FC = 3.1, pValue = 1,09E-168
#272 Ar = Upregulated, log2FC = 1.67, pValue = 1.6E-142

15. The text on p.8 and p.9 is quite dense, and there's no significance associated with any of the statements (although the methods suggest a hypergeometric test was done). For example, it's unclear how statements about replaced and joined are made as there's only one time point and there's no kinetic data.

R. Significance of hypergeometric tests is now indicated in the legend of Figure 3G. We understand the point raised by the Reviewer, terms like 'replaced' and 'joined' give a sense of a continuum that do not reflect the experimental setting. In the revised manuscript we changed 'replaced' with 'in place of' (*p*) and 'joined' with 'alongside'(*a*).

16. The introduction to FOXA1 mutations on p.10 would be better in the introduction

R. We thank the Reviewer for her/his suggestion. However, we would prefer to keep this part at the beginning of the of the results section focused on the role of FOXA1 mutations in prostate cancer. We retain that this brief description of the incidence and types of FOXA1 mutations in PCa and their functional roles in the tumorigenic process may help readers better understand the rationale underlying this part of the manuscript.

17. S4 - the nomenclature on the Venn diagrams is unclear to this reviewer

R. Apologize if the nomenclature on the Venn diagrams was not enough clear. Similar to Figure 3H, the two upper diagrams indicate respectively the upregulated (red, n=1830) and downregulated (purple, n=1173) differentially expressed genes (DEGs, FC>1.5; pValue <0.05) in the comparison ENRADA vs ERNAD-. The lower diagram represents the number of promoters (PE, Figure 3H) and distal elements (DE, Figure EV4) identified by ChIP-seq in Adams et al., 2019 to be bound by Foxa1 only -indicated as F1- (3H, n=1694; EV4B, n=4187), where Foxa1 was found in place of Ar -indicated as F1a(*p*)- (EV4C, n=685), where Foxa 1 was found together with Ar -indicated as F1a(*a*)+FA1- (EV4E, n=174), and, finally, where Ar was found alone -indicated as A1+A1a+A1b- (EV4G, n=52). The intersections between lower and upper diagrams indicate the numbers of up- or down-regulated genes having Foxa1 bound to a genomic region inside a window of ± 2.5 Kb from the gene (promoter elements/PE, Figure 3H), or Foxa1 and/or Ar bound to a genomic region in the proximity of the gene but outside the window of ± 2.5 Kb from the gene (distal elements/DE, Figure EV4 B,C,E, and G).

18. Unclear rationale for the study of the Fox mutations, as it doesn't obviously relate the RARg or ATRA signaling as developed in the previous experiments, and somewhat overlaps with the Adams et al paper. Apologies if this is misunderstood.

R. We understand the concerns raised by our Reviewer about the study of FOXA1 mutations associated with prostate cancer. The reason is mainly dependent by the fact that in presence of minimal RA signaling as for standard (ENRAD) growth conditions, expression of endogenous wild type Foxa1 is very low and differentiation of luminal progenitors shaping well-structured organoids reduced. This system can easily discriminate gain or loss of function mutations of Foxa1 through the expression of exogenous Foxa1 mutant proteins simply by tracking the generation of large hollow organoids. Moreover, as preliminary shown in this rebuttal (Reviewer #1, Point #3), loss of function mutations of Foxa1, such as F254E255, might be tested for their dominant negative role towards endogenous wild type Foxa1 by testing the ability of ATRA to rescue organoids formation. ATRA administration could offer an interesting matter of debate from a pre-clinical and clinical perspective as potential tool to counteract the oncogenic activity associated with loss-of-function mutations of FOXA1 in prostate and breast tumors as also suggested by Reviewer #3.

Referee #3:

In this manuscript, Dario De Felice and collaborators shed light on Retinoic acid (RA) as a key signal regulator for glandular identity in adult prostate progenitors and suggest that disruptions in vitamin A metabolism may be a risk factor for prostate diseases. The scientific article focuses on the critical roles of Vitamin A (retinol) in embryonic development, organogenesis, and tissue differentiation. It explores the mechanisms of action of Vitamin A, specifically through its active derivative, retinoic acid (RA), and its receptors (RARa, RARb, and RARg). The study highlights the significance of RA-RARg signaling in prostate development, particularly through the target gene Foxa1, which is essential for luminal lineage differentiation and proper prostate function.

Key findings include i) Foxa1's role in shaping androgen signaling and promoting the expression of genes critical for luminal progenitor cells; ii) Challenges in modeling the luminal compartment in vitro, with current organoid systems lacking robust secretion of kallikrein-rich fluids; iii) Identification of FOXA1 as a major target of RA signaling, with implications for prostate cancer, where FOXA1 mutations may enhance transcriptional activity and contribute to tumorigenesis. The study emphasizes the need for further research on the molecular mechanisms regulated by RA and FOXA1, particularly in the context of cancer treatment, where retinoid derivatives may offer therapeutic potential for tumors with dysfunctional RA signaling and specific FOXA1 mutations.

This study offers new insights into the signaling pathways and molecular circuits that regulate prostate progenitor commitment and adult tissue homeostasis. The article provides an extensive overview of Vitamin A's role in embryonic development, organogenesis, and tissue differentiation, highlighting the nutrient's critical importance in these processes. The identification and characterization of retinoic acid (RA) and its receptors (RARa, RARb, RARg) are thoroughly detailed, providing significant insights into their distinct expression patterns and functions. The use of mouse prostate organoids to study RA signaling and its impact on Foxa1 offers a novel and practical approach to understanding the molecular mechanisms underlying prostate development. Similarly, it paves the way for more precise assessments of retinoid derivatives in treating prostate tumors. Exploring FOXA1 mutations in prostate and breast cancers and their impact on epithelial transformation and tumorigenesis provides valuable implications for cancer research and potential

therapeutic strategies. Hence, this work will undoubtedly be of general interest to the readers of EMBO Reports.

R. We greatly appreciate the very positive comments of our Reviewer.

However, there are some minor weaknesses to consider that should be addressed before the publication of this manuscript:

1- While the use of organoid systems is innovative, the study acknowledges that these systems currently lack robust and reproducible secretion of kallikrein-rich fluids, limiting the model's physiological relevance. This issue should be more clearly stated in the discussion.

R. We thank our Reviewer for her/his suggestion. A novel paragraph addressing this point has been included in the Discussion of the revised manuscript.

“Although this work advances the ability to model the prostate luminal progenitor compartment in vitro, prostate organoid systems is still limited in the ability to efficiently generate fully differentiated luminal cells, partially restricting the physiological relevance of this model. We expect that continuous progress modeling prostatic functions in vitro will further extend our comprehension of the underlying molecular mechanisms, including the interplay between FOXA1 and AR transcriptional regulation.”

2- Also, the article points out that the molecular circuits underlying the processes influenced by RA signaling and Foxa1 remain understudied, indicating a gap in the comprehensive understanding of these pathways.

R. We completely agree with our Reviewer that more investigations will be necessary in the future to fully deconvolute how RA-Foxa1 molecular circuit governs the differentiation and homeostasis of the luminal progenitor compartment of the adult prostate. Understanding these processes could have very important implications in defining the risk factors that contribute to prostate tumorigenesis.

3- Although the study provides significant insights, the exact molecular mechanisms by which Foxa1 and RA signaling regulate luminal progenitor fate and prostate function are not fully elucidated. While Foxa1 is identified as a major target, the author's study may explore additional targets and pathways involved in RA signaling to provide a more holistic view of its role in prostate development and cancer.

R. We agree with our Reviewer that a more holistic view will better frame the role of RA signaling in prostate development and tumorigenesis. Indeed, new projects have been initiated based on integrated computational data from ChIP and RNA sequencing. In any case, these new studies will require a certain amount of time that cannot match the time scheduled for the revision.

4- Merge in Fig. 5(F) is advised.

R. We apologize with our Reviewer but there is no Figure 5F in the manuscript.

5- The article could strengthen its impact by outlining more specific future research directions or experimental approaches to address the identified gaps and weaknesses.

R. We thank the Reviewer for her/his suggestion. The following paragraphs of the original Discussion have been expanded with the underlined sentences in the revised manuscript.

“In cancer cells with loss-of-function mutations of FOXA1 that preserve the ability to form homodimers, concomitant induction of both wild-type and mutant alleles will presumably result in a dominant negative effect of the mutant protein on the regulation of DIV elements. In contrast, a benefit of FOXA1 overexpression should be expected on DIV controlled genes in the presence of loss-of-function mutant alleles unable to homodimerize, and on targeted genes where FOXA1 works as a monomer. Accordingly, preliminary experiments in mouse prostate organoids bearing the F254E255 mutation of Foxa1 showed the ability of ATRA to rescue the formation of large hollow organoids in vitro, likely caused by the induction of the endogenous Foxa1^{WT} alleles.”

“Noteworthy, subgroups of patients with superficial papillary or resected high-risk non-muscle invasive bladder cancer showed reduced recurrence rate and cancer progression (Sabichi et al. 2008; Studer et al. 1995; Alfthan et al. 1983), while few patients with advanced breast cancer achieved partial response or had stable disease (Sutton et al. 1997). In this scenario, orthotopic transplants of mPrOs (Cambuli et al., 2022) carrying tumor-associated Foxa1 mutations in syngeneic wild type adult mice will provide a valuable preclinical platform to test the efficacy of RA signaling in counteracting the tumorigenic process according to the specific class of Foxa1 mutations.”

6- A general review of abbreviations is advised (see pg.3 AR).

R. We thank our Reviewer for her/his advice. Abbreviations have been carefully reviewed in the revised version of the manuscript.

Dear Andrea,

Thank you for the submission of your revised manuscript. We have now received the enclosed reports from the referees and I am happy to say that both support its publication now. Only a few editorial requests will need to be addressed before we can proceed with the official acceptance of your manuscript.

- Your ms has 5 main figures and should thus be published as a short report with combined results and discussion sections. If you prefer not to combine these sections, please add one more main figure to the ms file.
- Please add up to 5 keywords to the ms file.
- Please rename "Data and code availability" to "Data Availability Section" and place it after the Methods, before Acknowledgments; please insert the PRJNA1064118 link provided in the Cover Letter in the DAS
- Please correct the conflict of interest subheading to "Disclosure Statement and Competing Interests"
- Please correct the name discrepancy - Srinivasaraghavan Kannan in the ms vs. Srinivasaragavan Kannan in eJP
- Please remove the author credits from the ms file. All credits are entered during online ms submission.
- All FUNDING INFO must be entered during online ms submission. Currently missing: European Regional Development Fund (ERDF) 2014-2020, Fondazione Trentina per la Ricerca sui Tumori (FTRT), core funding from the Department CIBIO, University of Trento (Starting Grants Young Researchers 2019) and the University of Trento (Ph.D. fellowship), Pezcoller Foundation (Ph.D. fellowship)
- Tables EV1-EV4 are datasets and they need to be uploaded and renamed as Dataset EV1-EV4 (source file names, titles, ms callouts need to be updated); their legends are provided in a separate file but we need them each inserted in their corresponding Excel file (as a separate sheet/tab)
- APPENDIX 1 FILE WITH Table of Content: in, but missing page numbers in the Table of Content on the title page; author list and affiliations are not needed (the title "Appendix" and the ToC with page numbers would be enough)
- The Methods section needs to include a Reagents and Tools Table (listing key reagents, experimental models, software and relevant equipment and including their sources and relevant identifiers). A downloadable template (.docx) for the Reagents and Tools Table can be found in our author guidelines: <<https://www.embopress.org/page/journal/14693178/authorguide#manuscriptpreparation>>.
- Please upload the source data as one (zipped) folder per figure. Each figure folder should have one file per panel.
- Materials and Methods should be Methods
- Inkscape and BioRender.com should be acknowledged at the end of the Methods section in the following way:
Graphics:
(some of the... OR Figure #... OR synopsis) Graphics were created with BioRender.com and Inkscape.
- Our routine image analysis found that Figure 2D is reused in Figure EV3B - but this is not listed in the figure legend. Figure 4E EV/ENRAD cell is reused in Figure EV3D but this is not in the legend. Please clarify/explain and correct.
- Please note that information related to n is missing in the legend of figure 2h.
- Please note that n=2 in figure 2g. If n=2 no statistics should be calculated. In this case, please show all individual data points along with their mean.
- Please note that the exact p values are not provided in the legends of figures 1d-e, h; 2h; 3g; EV 3c; EV 4a.
- Please indicate the statistical test used for data analysis in the legends of figures EV 4d, f, h.
- Please note that in figures 2h; 3g; there is a mismatch between the annotated p values in the figure legend and the annotated p values in the figure file that should be corrected.
- Please note that the scale bar needs to be defined for figures 2i.

- Please note that in figure EV 3d; the scale bar unit should be corrected from μM to μm (in the figure legend).
- Please note that the white arrowhead is not defined in the legend of figure EV 3e. This needs to be rectified.
- Please note that the data callouts in the text for ""GSE128867"", ""Crowley L, et al."" data citation does not include "Data ref:" as a prefix.

I would like to suggest a few minor changes to the abstract that needs to be written in present tense. Please let me know whether you agree with this:

Retinoic acid (RA) signaling is a master regulator of vertebrate development with crucial roles in body axis orientation and tissue differentiation, including in the reproductive system. However, a mechanistic understanding of how RA signaling governs cell lineage identity is often missing. Here, leveraging prostate organoid technology, we show that RA signaling orchestrates the commitment of adult mouse prostate progenitors to glandular identity, epithelial barrier integrity, and specification of prostatic lumen. RA-dependent RAR γ activation promotes the expression of Foxa1, which synergizes with the androgen pathway for luminal expansion, cytoarchitecture and function. FOXA1 mutations are common in prostate and breast cancers, though their pathogenic mechanism is incompletely understood. Combining functional genetics with structural modeling of FOXA1 folding and chromatin binding analyses, we discover that FOXA1F254E255 is a loss-of-function mutation compromising its transcriptional function and luminal fate commitment of prostate progenitors. Overall, we define RA as an instructive signal for glandular identity in adult prostate progenitors. Importantly, we identify cancer-associated FOXA1 indels affecting residue F254 as loss-of-function mutations promoting dedifferentiation of adult prostate progenitors.

I also slightly modified your short summary and bullet points. Do you agree with:

Leveraging mouse adult prostate organoids reveals molecular circuits regulating prostatic luminal lineage commitment. The androgen pathway together with retinoic acid/RAR γ /Foxa1 signaling programs luminal progenitor identity and this is dysregulated in the context of Foxa1 recurrent oncogenic mutations.

- Nanomolar amounts of all-trans retinoic acid (ATRA) triggers luminal fate and lumenogenesis in mouse prostate organoids.
- RA-RAR γ signaling promotes Foxa1 expression in adult prostate progenitors.
- Foxa1 induces luminal progenitors genes and structural protein expression.
- Recurrent prostate cancer indel mutation of Foxa1 F254E255 shows impaired pro-luminal activity and reduced DNA binding stability.

Best regards
Esther

Referee #1:

The manuscript has improved and points were addressed, no further comments

Referee #2:

The authors have really undertaken a very strong revision of a manuscript, which was already compelling, and been highly responsive to quite diverse questions from the reviewers. I strongly recommend this manuscript to be accepted

All editorial and formatting issues were resolved by the authors.

Prof. Andrea Lunardi
University of Trento, Italy
CIBIO
Via Sommarive 9 (Povo)
Trento, Trento 38123
Italy

Dear Andrea,

I am very pleased to accept your manuscript for publication in the next available issue of EMBO reports. Thank you for your contribution to our journal.
